# DiEP: Adaptive Mixture-of-Experts Compression through Differentiable Expert Pruning

**Sikai Bai**
HKUST
Hong Kong, China
sbaiae@connect.ust.hk

**Haoxi Li**
HKUST
Hong Kong, China
hligb@connect.ust.hk

**Jie Zhang** [*]
HKUST
Hong Kong, China
csejzhang@ust.hk

**Zicong Hong**
HKUST
Hong Kong, China
congcong@ust.hk

**Song Guo** [*]
HKUST
Hong Kong, China
songguo@cse.ust.hk

## Abstract

Despite the significant breakthrough of Mixture-of-Experts (MoE), the increasing scale of these MoE models presents huge memory and storage challenges. Existing MoE pruning methods, which involve reducing parameter size with a uniform sparsity across all layers, often lead to suboptimal outcomes and performance degradation due to varying expert redundancy in different MoE layers. To address this, we propose a non-uniform pruning strategy, dubbed **Di**fferentiable **E**xpert **P**runing (**DiEP**), which adaptively adjusts pruning rates at the layer level while jointly learning inter-layer importance, effectively capturing the varying redundancy across different MoE layers. By transforming the global discrete search space into a continuous one, our method handles exponentially growing non-uniform expert combinations, enabling adaptive gradient-based pruning. Extensive experiments on five advanced MoE models demonstrate the efficacy of our method across various NLP tasks. Notably, **DiEP** retains around 92% of original performance on Mixtral $8\times7$B with only half the experts, outperforming other pruning methods by up to 7.1% on the challenging MMLU dataset.

## 1 Intorduction

Large Language Models (LLMs), such as GPT4 [36] and Llama series [14, 33], have demonstrated remarkable performance across diverse domains. However, real-world deployment poses significant challenges due to an ever-growing number of parameters, including high computational demands and storage costs. To address these issues, the Mixture-of-Experts (MoE) architecture [10, 12, 41] has emerged as a promising solution, activating only a subset of parameters during training and inference. Notable MoE-based models, such as Mixtral $8\times7$B [21], and DeepSeek V3 [27], achieve faster inference while maintaining competitive performance with dense models [14, 3] of comparable scale. Despite their computational efficiency, MoE models suffer from substantial memory and storage costs due to larger model sizes, making their deployment in resource-constrained environments challenging [19, 2]. For example, DeepSeek V3 has 256 experts per layer and 671B parameters.

Recent empirical analyses have shown that the routing policies learned by current MoE LLMs yield markedly unbalanced expert utilization [7, 28]. To mitigate the attendant waste of parameters, a growing body of work aims to prune experts while preserving the task performance of the full MoE model.

---

[*]Corresponding authors

39th Conference on Neural Information Processing Systems (NeurIPS 2025).

Most existing approaches impose a uniform sparsity budget on each layer: they either drop a fixed number of experts in each layer, or exhaustively search the combinatorial space of per-layer expert subsets. For instance, Zhang et al., [49] remove the same number of experts in each layer using activation-frequency heuristics, whereas search-based methods such as EEP [29] and NAEE [32] enumerate all $k$-expert combinations inside each MoE layer. Unfortunately, considering the discrepancy of expert redundancy across different MoE layers (i.e., more number of experts are required to be activated in shadow layers than deeper layers, as demonstrated in Sec. 5.3.2), simply applying uniform pruning ratio for all layers may cause poor performance during inference. Worse still, such limitation cannot be solved through global layer-wise brute-force searches. For instance, in a 64-expert layer, pruning only $12.5\%$ of the experts ($k = 8$) already requires evaluating $\binom{64}{8} \approx 4 \times 10^8$ configurations, making exhaustive global optimization computationally infeasible.

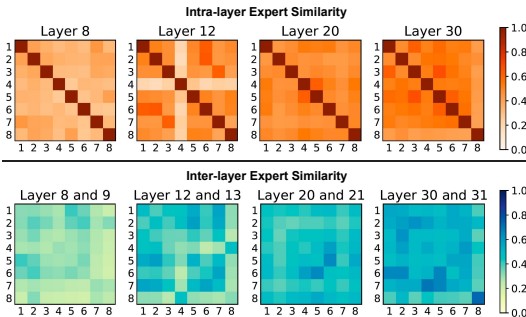

Figure 1: Visualized analysis of the intra-layer and inter-layer similarity between expert pairs for different MoE layers in Mixtral $8\times7$B through RBF kernel-based CKA criteria [22]. Darker colors represent higher expert similarity.

Layer-aware strategies have begun to surface, but they still fall short of capturing the heterogeneous relationships between layers. Among them, Li et al. [24], merge infrequently routed experts into their high-traffic counterparts after within-layer normalization of activation counts. Such normalization, however, erases cross-layer information and implicitly assumes that redundancy is independent across depth. Figure 1 contradicts this assumption: (i) the intra-layer similarity matrices for layers 8, 12, 20, and 30 exhibit distinct block structures and sparsity patterns, and (ii) the inter-layer CKA heatmaps reveal both strongly correlated and strongly divergent expert pairs across adjacent layers. These observations underscore the need for an adaptive, depth-sensitive pruning framework that leverages both intra- and inter-layer statistics to decide how many and which experts to retain at each layer.

To tackle these obstacles, we propose a novel and efficient approach, **Di**fferentiable **E**xpert **P**runing (**DiEP**), which reformulates expert pruning as a continuous optimization problem. Specifically, instead of searching over a global discrete search space with exponentially increasing choices, DiEP relaxes expert selection into a differentiable process. It performs joint optimization to determine the relative significance of experts within each layer and inter-layer importance scores that regulate the contribution of different layers. By incorporating layer-aware importance modulation, DiEP enables a globally optimized selection of experts through gradient-based optimization, effectively capturing both expert-level and layer-level impacts on the pruning process. Beyond permanently eliminating unimportant experts, we further propose an online expert skipping mechanism that assigns decayed expert weights to highly similar experts during inference. It bypasses redundant expert computations for each input token and accelerates inference speed.

While the idea of continuous search has been explored in dense neural architectures [11, 30, 43, 48], DiEP first introduces this principle into the sparsely activated MoE paradigm—a setting with fundamentally different structural and computational constraints. Unlike traditional bilevel differentiable search methods [30], DiEP jointly optimizes intra-layer expert logits and inter-layer importance scores in a single-stage training process, guided by a lightweight reconstruction regularizer and without reliance on a validation set. By decoupling gradient updates for intra- and inter-layer importance scores, DiEP mitigates optimization interference and enables a global ranking mechanism that produces precise, depth-aware sparsity patterns without manual heuristics. Extensive experiments show that DiEP outperforms other pruning methods in various NLP tasks and MoE architectures, while reducing model size and enhancing inference efficiency.

## 2 Related Work

### 2.1 Sparse Mixture-of-Experts Models (SMoE)

It selectively activates a small subset of specialized networks (experts) for each input, enabling efficient model scaling [5, 20]. In early research, Shazeer et. al. [41] introduced the Sparsely-Gated

MoE layer, demonstrating the effectiveness of selective expert activation. 23 advanced SMoE by implementing a distributed architecture that enabled efficient scaling across multiple devices. Recent studies have further refined SMoE architecture based on SOTA LLMs [44]. Mixtral models [21] demonstrated successful scaling with a balanced approach of using two experts per token; Qwen-MoE [44] and DeepSeek-MoE [10, 16] explored larger expert pools with selective activation. They have attracted great attention from the AI community. Despite these advances, current SMoE-LLM architectures require huge memory to load trillion parameters and suffer from low expert utilization during inference.

## 2.2 Expert Pruning for SMoE

Inspired by recent advances in LLMs [31, 25], expert pruning has become a promising technique to reduce model complexity while maintaining performance for SMoE. Existing solutions can be divided into two branches: 1) *Features statistics* identifies unnecessary experts based on the activation frequency or feature similarity, but such methods either dramatically compromise performance [50, 35] or rely on post-processing [24, 15]. 2) *Greedy search* heuristically searches all possible choices for pruned experts within each layer, which becomes impractical for the latest SMoE models due to exhaustive search [32] or task-specific fine-tuning [29, 45]. To make matters worse, all the above methods either fail to account for the varying levels of expert redundancy in different MoE layers by applying identical pruning rates or incur heavy computation costs to implement non-uniform pruning. However, our DiEP uses parameter-efficient intra-layer and inter-layer differentiable optimization to adaptively search pruned experts, reducing redundancy based on each layer/expert characteristics while keeping the full model's performance.

## 2.3 Continous Optimization

The concept of architecture search and optimization within a continuous domain has been explored before [1, 30, 39, 40, 43, 48]. Early research [1, 40] focuses on fine-tuning architectural components such as filter shapes or branching patterns in convolutional networks. After that, a representative framework DARTS [30] and its variants [6, 46] were introduced to learn high-performance architecture building blocks with complex graph topologies, but they employ memory-intensive operations in the architectural search process and require costly nested optimization and validation-set dependence. Moreover, DiffPruning [39] was proposed to remove redundant parallel processing units in dense transformer architectures through differentiable pruning. Although it updates head importance scores and model parameters via monotonic gradient descent, there's a risk of gradient conflict between importance scores and weight matrices, and it requires threshold tuning after continuous relaxation. In contrast, our DiEP method decouples the intra-layer and inter-layer gradient optimization paths and achieves exact sparsity through a unified global ranking without threshold tuning. To the best of our knowledge, our DiEP is the first method to explore continuous expert search for Mixture of Experts (MoE) architectures in the context of Large Language Models.

## 3 Preliminary: Mixture-of-Experts (MoE) Language Model

Generally, a Mixture-of-Experts (MoE) model consists of $L$ layers, where each layer $l$ ($l = 1, \ldots, L$) contains $N$ experts. The input to all experts in the $l$-th layer is denoted as $\boldsymbol{x}^{(l)} \in \mathbb{R}^d$, where $d$ is the input dimension. A router network produces routing logits $\zeta_i^{(l)}$ for each expert $i$ ($i = 1, \ldots, N$), which are normalized using a softmax function to compute the routing weights $w_i^{(l)}$:

$$w_i^{(l)} = \frac{\exp(\zeta_i^{(l)})}{\sum_{j=1}^{N} \exp(\zeta_j^{(l)})}, \tag{1}$$

where $w_i^{(l)}$ represents the contribution of expert $i$ in layer $l$.

To enforce sparsity, the router network selects the top-$k$ experts with the largest routing weights $w_i^{(l)}$. The output of the $l$-th MoE layer is then computed as:

$$\boldsymbol{y}^{(l+1)} = \sum_{i \in \text{Top-}k(w^{(l)})} w_i^{(l)} \cdot \text{FFN}_i(\boldsymbol{x}^{(l)}), \tag{2}$$

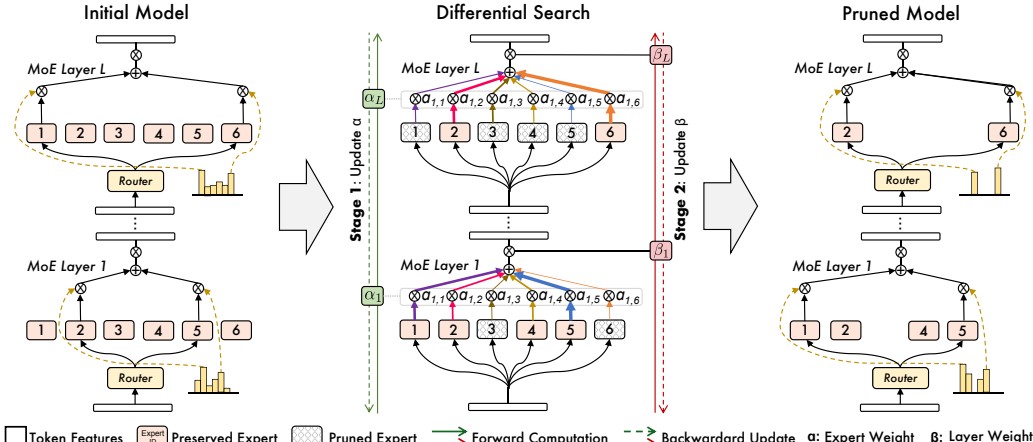

Figure 2: The schematic illustration of the Differentiable Expert Pruning (DiEP) Framework. (a) Initial MoE model with substantial expert redundancy and memory cost. (b) Differentiable Pruning: transforming discrete expert search into a continuous optimization by jointly learning intra-layer expert scores ($\alpha$) and inter-layer importance ($\beta$) via an alternating update strategy, enabling adaptive non-uniform pruning. (c) Final pruned model: achieving a streamlined MoE architecture that maintains high performance while reducing the model's footprint.

where $\text{FFN}_i(\cdot)$ denotes the feed-forward function of expert $i$, and Top-$k(w^{(l)})$ refers to the indices of the $k$-largest routing weights. The final output $\boldsymbol{y}^{(l+1)}$ is passed to the subsequent layer.

## 4 Method

### 4.1 Sparse Expert Search Space

Following the design principles of differentiable architecture search, we first define a sparse expert search space tailored for Mixture-of-Experts (MoE) architectures, as illustrated in Figure 2. In this framework, an MoE layer is modeled as a directed acyclic graph (DAG) consisting of only two nodes: an input node representing the token representations entering the expert layer and an output node representing the sum of selected expert transformations. Instead of treating individual experts as independent computational units, we formulate the expert pruning process as a discrete operation over a single aggregated expert node.

Based on expert pruning principles, a subset of experts is retained according to their importance, governed by a binary selection mask $m_i^{(l)} \in \{0, 1\}$, where $m_i^{(l)} = 1$ indicates that expert $i$ is retained, and $m_i^{(l)} = 0$ indicates pruning. The expert aggregation process in an MoE layer is then expressed as:

$$\boldsymbol{y}^{(l+1)} = \sum_{i=1}^{N} (m_i^{(l)} \cdot \text{FFN}_i)(\boldsymbol{x}^{(l)}), \tag{3}$$

where $\text{FFN}_i(\cdot)$ denotes the feed-forward function of expert $i$.

This discrete selection process inherently results in a non-differentiable search space, making direct optimization intractable. To enable gradient-based optimization and structured pruning within the MoE framework, we introduce a continuous relaxation mechanism, allowing smooth updates to the expert selection process while preserving the structured sparsity of the model.

### 4.2 Continuous Relaxation and Optimization

Specifically, we decompose the expert importance into two components: *intra-layer importance scores* $\alpha$ that determine the relative significance of experts within each layer and *inter-layer importance scores* $\beta$ that regulate the contribution of different layers in the selection process. This formulation allows us to perform structured pruning in a data-driven and globally optimized manner.

We define the intra-layer importance weights, $\bar{\alpha}_i^{(l)}$, by normalizing the intra-layer importance scores $\alpha_i^{(l)}$ using a softmax function:

$$\bar{\alpha}_i^{(l)} = \frac{\exp(\alpha_i^{(l)})}{\sum_{j=1}^{N} \exp(\alpha_j^{(l)})}, \tag{4}$$

where $\alpha_i^{(l)}$ are learnable logits that determine the relative importance of experts within layer $l$. This normalization ensures a smooth and differentiable selection process. Similarly, the inter-layer importance score $\beta^{(l)}$ is introduced as a trainable scalar that modulates the overall contribution of layer $l$. The output of an MoE layer $l$ is then computed as:

$$\boldsymbol{y}^{(l+1)} = \beta^{(l)} \sum_{i=1}^{N} \bar{\alpha}_i^{(l)} \cdot \text{FFN}_i(\boldsymbol{x}^{(l)}). \tag{5}$$

To ensure that the pruned model retains fidelity to the original MoE model $\mathcal{F}(\boldsymbol{x})$ (before pruning), we introduce a reconstruction regularization term $\Phi(\alpha, \beta)$, defined as:

$$\Phi(\alpha, \beta) = \|\mathcal{F}'(\boldsymbol{x}; \alpha, \beta) - \mathcal{F}(\boldsymbol{x})\|_F, \tag{6}$$

where $\|\cdot\|_F$ denotes the Frobenius norm. This regularization encourages the pruned model $\mathcal{F}'$ to maintain consistency with the original model.

The overall objective function is formulated as:

$$\min_{\alpha, \beta} \mathcal{L}(\alpha, \beta) := \mathcal{L}_{ce}\big(\boldsymbol{y}, \mathcal{F}'(\boldsymbol{x}; \alpha, \beta)\big) + \lambda \Phi(\alpha, \beta), \tag{7}$$

where $\lambda$ is a regularization coefficient, and $\mathcal{L}_{ce}$ is the cross-entropy loss.

**Alternating Update Strategy.** To optimize the objective function, we adopt an alternating update strategy where the intra-layer importance scores $\alpha$ and inter-layer importance scores $\beta$ are updated iteratively:

$$\alpha^t \leftarrow \alpha^t - \eta_\alpha \nabla_\alpha \mathcal{L}(\alpha^t, \beta^t), \tag{8}$$

$$\beta^t \leftarrow \beta^t - \eta_\beta \nabla_\beta \mathcal{L}(\alpha^t, \beta^t). \tag{9}$$

Here, $t$ denotes the iteration index, $\eta_\alpha$ and $\eta_\beta$ are the learning rates for $\alpha$ and $\beta$, respectively, and $\mathcal{L}(\alpha, \beta)$ represents the overall objective function defined in Equation 7. From the theoretical perspective, we summarize the optimization process in Algorithm 1 and provide the detailed convergence analysis in Appendix B.2.

**Pruning Strategy.** To derive a discrete architecture, we apply a structured pruning mechanism that eliminates the least significant experts based on their global contribution across all layers. Instead of pruning experts layer-by-layer in isolation, we leverage the learned intra-layer importance scores $\alpha_i^{(l)}$ and inter-layer importance scores $\beta^{(l)}$ to determine expert significance in a unified manner.

Formally, the overall importance of expert $i$ in layer $l$ is computed as the product of its intra-layer and inter-layer importance scores:

$$s_i^{(l)} = \alpha_i^{(l)} \cdot \beta^{(l)}. \tag{10}$$

Given the expert sparsity ratio $r$, the total number of experts to be pruned across the entire MoE model is $K = NLr$, where $N$ is the number of experts per layer and $L$ is the number of layers. The pruning process is performed by globally sorting all experts based on their importance scores $s_i^{(l)}$ and removing the bottom-$K$ least significant experts. The resulting pruning mask $m_i^{(l)}$ is defined as:

$$m_i^{(l)} = \begin{cases} 0 & \text{if } i \in P, \\ 1 & \text{otherwise,} \end{cases} \tag{11}$$

where $P$ is the set of the bottom-$K$ experts selected for pruning.

By jointly considering both intra-layer and inter-layer importance scores, this pruning strategy ensures a globally optimized selection of experts, effectively reducing computational redundancy while maintaining structural balance across layers.

### 4.3 Adaptive Skipping During Inference

During the inference process, processing each token with all selected top-$k$ experts introduces unnecessary computational overhead, but researchers in [32] find that not every selected expert provides essential contributions for tokens. This observation motivates the need for adaptive expert skipping, which selectively bypasses less significant experts during inference to enhance efficiency. For each token $x$ in an MoE layer, the top-$k$ experts are chosen using routing weights $w = \{w_{e_0}, w_{e_1}, \ldots, w_{e_{k-1}}\}$, and their outputs are denoted as $y_{e_0}, y_{e_1}, \ldots, y_{e_N}$. Following common practice, we assume $k = 2$ for simplicity. Unlike previous approaches [32] that rely solely on routing weights, our method incorporates expert similarity to dynamically skip less important experts during inference, thereby enhancing computational efficiency.

Assume experts with indices $e_0$ and $e_1$ are selected, with $w_{e_1} < w_{e_0}$. To improve inference speed, if $w_{e_1} < \gamma w_{e_0}$, expert $e_1$ is skipped, where $\gamma$ is a hyperparameter specific to each MoE layer and generation step.

In our implementation, $\gamma$ is calculated as the product of two factors. First, $\gamma_1$ is determined as the median value of $\frac{w_{e_1}}{w_{e_0}}$ across sampled calibration data for each MoE layer. Second, $\gamma_2$ is computed based on the similarity between expert outputs, evaluated using Centered Kernel Alignment (CKA) [22]. Specifically, $\gamma_2$ is the ratio of the CKA similarity $\rho(y_{e_0}, y_{e_1})$ to the mean CKA similarity $\rho(y_{e_i}, y_{e_j})$ across all data samples in layer $l$. The final value of $\gamma$ is given by:

$$\gamma = \gamma_1 \times \gamma_2. \tag{12}$$

This method dynamically adjusts expert skipping based on both expert routing weights and similarity, significantly enhancing inference efficiency and maintaining model performance. In our experiments, we observe a speedup in inference $1.2\times$ to $1.3\times$ while retaining approximately 92% of the average performance with only half of the experts on Mixtral $8\times$7B.

## 5 Experiments

### 5.1 Experimental Settings

**Model Settings.** Our primary experiments are conducted using the widely adopted SMoE model, Mixtral $8\times$7B. To validate our method's generalizability across different models, we extend our experiments to an instruction-following model, Mixtral $8\times$7B-Instruct, the larger model Mixtral $8\times$22B, and other types of SMoE models such as Deepseek-MoE-16B and Qwen2-57B-14A. In the Mixtral architecture, each token activates two experts in every MoE layer. Both Mixtral $8\times$7B and Mixtral $8\times$7B-Instruct contain 32 sparse MoE layers with eight experts per layer while Mixtral $8\times$22B contains 56 MoE layers with the same number of experts per layer. Deepseek-MoE-16B employs a different architecture with 28 layers and 64 experts per layer, where each token passes through two shared experts and selects six additional experts. Similarly, Qwen2-57B-14A consists of 28 MoE layers with 64 experts in each layer but utilizes eight experts per token during inference.

**Dataset.** We evaluate model performance using the Language Model Evaluation Harness library [13] across four zero-shot tasks: MMLU [18], OpenBookQA [34], BoolQ [8], and RTE [4]. MMLU [18] represents the most comprehensive and challenging benchmark, encompassing 57 subtasks distributed across four major domains: humanities, social sciences, STEM, and other. More results on other tasks are provided in the Appendix A.10.

**Implementation Details.**

During the expert pruning phase, we construct a small calibration subset with 128 samples from the C4 dataset for fine-tuning purposes. We implement parameter-efficient differential learning through alternating training cycles, with a 3:1 ratio between intra-layer scores $\alpha$ and inter-layer scores $\beta$ updates. Both training processes employ a learning rate of 5e-3 with a cosine learning rate scheduler. In addition, the complete training protocol consists of 10 epochs with a batch size of 16. For weight hyperparameter settings, we use $\lambda = 0.01$ for all Mixtral architectures and $\lambda = 0.01$ for other MoE models. All experimental evaluations are conducted using four NVIDIA GeForce A800 GPUs.

**Baselines.** We compare our method with the following pruning methods: *M-SMoE* [24], which merges experts based on customized permutation alignment and routing strategies; *Expert Trimming* [17],

Table 1: Zero-shot performance comparison of different expert pruning methods on Mixtral-8×7B, Mixtral-8×7B-Instruct, and Mixtral-8×22B. Expert sparsity $r$ indicates the proportion of pruned experts in the full model across all layers. The first and second columns represent results for expert sparsity $r = 25\%$ and $r = 50\%$, respectively.

| Model | Method $r = 25\%/50\%$ | MMLU humanities | social science | stem | other | avg | BoolQ | OpenBookQA | RTE | Average |
|---|---|---|---|---|---|---|---|---|---|---|
| Mixtral 8×7B | Full | 60.5 | 77.8 | 58.9 | 74.2 | 67.9 | 85.3 | 35.4 | 71.5 | 65.1 |
| | M-SMOE | 51.8/24.8 | 60.5/26.5 | 46.9/24.7 | 60.5/25.0 | 54.9/25.3 | 82.6/39.9 | 32.0/11.6 | 70.4/50.9 | 60.0/31.9 |
| | Expert Trimming | 49.2/36.9 | 59.7/45.6 | 45.0/35.1 | 58.2/43.4 | 54.1/45.7 | 77.2/76.6 | 33.0/26.4 | 56.6/55.9 | 55.2/51.2 |
| | NAEE | 52.4/43.5 | 66.4/52.7 | 49.0/40.4 | 63.7/43.5 | 58.7/47.3 | 84.0/80.8 | 32.6/28.8 | 67.9/61.4 | 60.8/54.6 |
| | S-MOE | 56.0/48.0 | 73.1/57.0 | 52.4/43.3 | 68.2/54.6 | 59.9/50.8 | 86.4/83.3 | 31.4/26.2 | 69.3/67.1 | 61.5/55.9 |
| | **DiEP(Ours)** | **58.8/52.9** | **75.4/69.3** | **56.8/49.1** | **72.0/63.5** | **64.9/57.9** | **86.6/84.0** | **33.1/29.6** | **70.7/68.2** | **63.8/59.9** |
| Mixtral 8×7B -Instruct | Full | 61.2 | 79.7 | 59.6 | 75.8 | 68.1 | 88.5 | 36.6 | 72.2 | 66.4 |
| | M-SMOE | 48.5/33.8 | 62.3/37.5 | 44.0/33.8 | 55.3/35.4 | 52.0/35.0 | 85.3/77.6 | 29.0/26.4 | 67.5/61.8 | 58.5/50.2 |
| | Expert Trimming | 52.9/45.0 | 74.3/61.1 | 50.5/39.2 | 64.8/50.8 | 58.6/47.3 | 86.3/83.0 | 37.0/32.3 | 63.2/66.8 | 61.3/57.3 |
| | NAEE | 55.9/48.7 | 69.5/55.6 | 54.1/42.3 | 68.7/56.2 | 62.4/52.8 | 87.3/84.8 | 35.6/30.4 | 70.0/**75.5** | 63.8/60.9 |
| | **DiEP(Ours)** | **61.2/55.1** | **78.1/72.3** | **59.4/53.4** | **73.8/67.8** | **67.3/61.3** | **87.7/85.6** | 35.9/31.0 | **72.2**/74.0 | **65.8/63.0** |
| Mixtral 8×22B | Full | 68.6 | 84.1 | 67.1 | 78.7 | 72.6 | 87.9 | 35.8 | 71.5 | 67.0 |
| | M-SMOE | 27.3/22.7 | 25.4/25.8 | 24.4/24.0 | 27.9/23.4 | 26.4/23.9 | 62.8/62.7 | 12.8/13.0 | 54.2/49.5 | 39.1/37.3 |
| | Expert Trimming | 58.0/45.7 | 74.9/57.7 | 54.1/42.0 | 70.2/45.7 | 64.3/47.8 | 81.5/74.4 | 35.2/27.0 | 69.3/57.4 | 62.6/51.7 |
| | NAEE | 60.4/53.9 | 78.0/67.2 | 59.5/52.3 | 73.0/64.2 | 67.7/59.4 | 87.4/80.5 | 35.0/31.1 | 70.1/67.9 | 65.1/59.7 |
| | S-MOE | 62.3/57.8 | 78.5/69.7 | 60.2/51.3 | 73.4/64.2 | 68.6/60.8 | 87.6/83.1 | **35.8**/33.2 | 71.1/68.1 | 65.7/61.3 |
| | **DiEP(Ours)** | **65.0/58.9** | **81.8/73.2** | **63.2/54.2** | **76.0/68.7** | **70.7/62.4** | **87.7/84.5** | 35.8/**34.4** | **71.3/70.4** | **66.4/62.9** |

(a) Deepseek-MoE-16B.     (b) Qwen2-57B-14A.

Figure 3: Zero-shot performance comparison on Deepseek-MoE-16B and Qwen2-57B-14A.

which removes less important experts using activation frequency or removes structured modules through layer and block dropping; *NAEE* [32], which enumerates expert combinations and selects optimal remaining experts by minimizing reconstruction loss; *S-SMOE* [49], which identifies and addresses expert redundancy through similarity-based pruning and merging operations.

## 5.2 Main results

To illustrate the efficacy of our proposed method, we report the performance comparison of DiEP and other state-of-the-art pruning methods through comprehensive experiments on five SMoE architectures across various tasks.

### 5.2.1 Results on Mixtral Models

Table 1 shows experimental results on Mixtral 8×7B, Mixtral 8×7B-Instruct, and Mixtral 8×22B, where all three architectures have 8 experts in each MoE layer and we prune them under 25% and 50% expert sparsity, respectively. ***Mixtral 8×7B***: Compared to other pruning strategies, our proposed DiEP significantly outperforms them on all tasks with a clear margin performance improvement, up to 7.1%. Specifically, when evaluated on MMLU, which is a challenging dataset with numerous sub-tasks, other methods suffer from performance bottlenecks under 50% expert sparsity, but our DiEP effectively alleviates the negative influence of removing a large number of experts.

These results demonstrate that DiEP effectively preserves the key expert knowledge by differential optimization and search on task-agnostic data using intra-layer scores and inter-layer scores. ***Mixtral 8×7B-Instruct***: our DiEP significantly surpasses other pruning strategies by a substantial margin. Specifically, DiEP achieves optimal performance with an average reduction of only 0.6% compared to the full model under 25% expert sparsity (i.e., removing 64 experts after pruning). These outcomes indicate that DiEP successfully mitigates the detrimental effects of expert pruning. ***Mixtral 8×22B***:

We further extend our pruning strategy to a larger model, Mixtral 8×22B which activates 39 billion parameters out of a total of 141 billion. Our proposed DiEP method continues to demonstrate substantial improvements across all tasks, retaining 94% of the full model's performance even after the removal of 50% of the experts. These results reveal the significant redundancy present in the MoE layers and showcase the scalability of DiEP for large-scale SMoE models.

### 5.2.2 Results on Deepseek and Qwen Models

To demonstrate the generalizability of our proposed method across various models, we further apply it to the Deepseek-MoE-16B and Qwen2-57B-14A architectures, which differ significantly from those Mixtral models. In these architectures, each layer comprises 64 experts, with 8 experts activated for each token at every layer. Specifically, as shown in Figures 3a and 3b, we averagely reduce the number of experts in each layers from 64 to 62, 60, 58, 56, 54, 52, 50 and 48 in both models.

***Deepseek-MoE-16B***: We observe that the frequency-based method (M-SMoE) suffers a significant performance degradation on the MMLU dataset. In contrast, our DiEP consistently showcases superior performance across various pruning ratios, achieving an average advantage of approximately 1.57% compared to second runner strategy (S-MoE), which relies on additional expert merging and large similarity matrix computations.After removing 244 experts from the full Qwen-MoE model, our DiEP retains a promising average performance of 68.7%, reflecting a mere 0.4% degradation compared to the full model.

***Qwen2-57B-14A***: Actually, DiEP always achieves comparable performance to the full model and surpasses all baseline methods across various tasks. This underscores the adaptability and effectiveness of our method for different SMoE models, grounded in general-purpose differential optimization.

### 5.3 Ablation Studies

#### 5.3.1 Effectiveness of Components

To measure the importance of key components in our DiEP, we conducted ablation studies on Mixtral 8×7B with the following variants. As shown in Table 2, ***Row 1*** serves as the baseline (NAEE), it only performs a layer-wise search for all possible expert combinations and has poor performance. ***Row 2*** focuses on learning intra-layer expert importance $\alpha$ to

Table 2: Performance analysis of different components.

| Method | MMLU | BoolQ | OpenBookQA | RTE | Avg. |
|---|---|---|---|---|---|
| Baseline | 58.7/47.3 | 84.0/80.8 | 32.6/28.8 | 67.9/61.4 | 60.8/54.6 |
| $W_\alpha$ | 60.5/51.0 | 86.0/82.8 | 32.2/27.8 | 67.5/65.3 | 61.6/56.7 |
| $W_\beta$ | 61.0/51.4 | 85.1/83.3 | 32.0/29.6 | 67.3/66.2 | 61.3/57.6 |
| $W_\alpha + W_\beta(random)$ | 57.6/49.2 | 85.6/83.4 | 32.3/27.2 | 66.4/62.1 | 60.47/55.5 |
| $W_\alpha + W_\beta(1:2)$ | 55.1/46.2 | 81.5/77.4 | 30.6/26.8 | 66.4/64.2 | 57.8/54.2 |
| $W_\alpha + W_\beta(2:1)$ | 63.3/54.2 | 85.4/83.5 | 32.6/29.8 | 68.2/67.5 | 62.4/58.8 |
| $W_\alpha + W_\beta(3:1)$ | 64.6/55.2 | 85.9/84.2 | 32.8/29.6 | 69.7/67.8 | 63.3/59.2 |
| **DiEP(Ours)** | **64.9/57.9** | **86.6/84.0** | **33.1/29.6** | **70.7/68.2** | **63.8/59.9** |

measure the global contribution of each expert. ***Row 3*** denotes the variant of eliminating $\alpha$, and applies inter-layer scores $\beta$ to reweight expert activation frequencies as global importance scoring of each expert. Compared with the baseline, the substantial performance gains demonstrate that the two components are both effective. To further investigate the efficiency of learnable inter-layer importance scoring, we replaced learnable $\beta$ with fixed scores in ***Row 4-7***, where a ratio of 2:1 means $\beta = 2$ for layers 1-16 and $\beta = 1$ for layers 17-32. The results show that assigning higher scores to lower layers yields better performance in Mixtral 8×7B, but this method of artificially fixing parameters cannot be generalized to other SMoE architectures and could result in a significant waste of computational resources to identify the optimal $\beta$. Furthermore, our method leverages complementary knowledge from intra-layer scores $\alpha$ and inter-layer scores $\beta$ for better expert selection, yielding superior performance.

#### 5.3.2 Visualization Analysis for $\alpha$ and $\beta$

To further validate the effectiveness of our proposed method, we visualized the variation of the updated intra-layer scores $\alpha$ and inter-layer scores $\beta$ after the pruning stage. As shown in Figure 4a, the distribution of intra-layer importance scores $\alpha$ reveals that experts in layers 1–15 tend to have higher average scores compared to those in layers 16–32. This suggests that shallower layers generally play a more significant role in the overall model. Figure 4b illustrates the inter-layer importance scores, which corroborate the intra-layer observations. The overall trend indicates that the alternating update strategy effectively captures both intra- and inter-layer dependencies, ensuring that the MoE

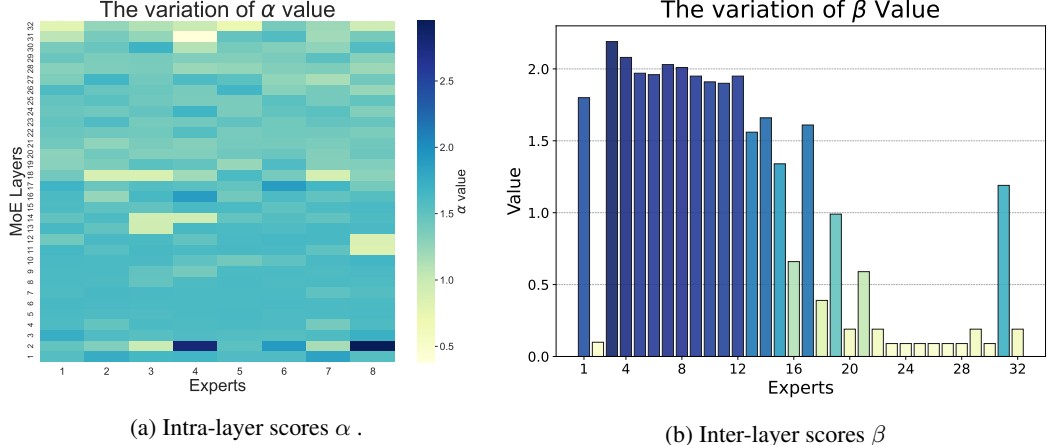

(a) Intra-layer scores $\alpha$ .        (b) Inter-layer scores $\beta$

Figure 4: Visualization of values distribution for intra-layer scores $\alpha$ and inter-layer scores $\beta$ on Mixtral 8×7B when $r = 50\%$..

model retains critical information from shallower layers. Furthermore, a closer examination of layer 2 reveals that two experts. Specifically, the fourth and eighth experts exhibit markedly high $\alpha$ values relative to their peers. This shows that these experts are consistently considered highly important by the model. Conversely, the remaining experts in layer 2 generally have low importance scores, indicating that the pruning strategy in this layer is not governed by a single importance criterion. Instead, it shows a clear preference for retaining these two experts through a global selection. Overall, these empirical findings further confirm the efficacy of our proposed differentiable expert pruning approach and underscore the synergistic relationship between $\alpha$ and $\beta$.

### 5.3.3 Computation Cost Analysis

We further analyze the efficiency of our DiEP during the pruning and inference stages. For pruning, as shown in Table 3, our baseline (NAEE), using an exhaustive heuristic search, becomes computationally prohibitive for models with large expert pools like Deepseek-MoE-16B and Qwen2-57B-14A. In contrast, our DiEP, with only a 0.01% parameter overhead, maintains consistent pruning time and achieves superior performance regardless of model architecture or expert count. Furthermore, Table 4 shows DiEP's inference cost reductions on Mixtral 8×7B in terms of latency and GPU

Table 3: Pruning time comparison of our DiEP and NAEE on different models under 25% expert sparsity.

| Method | Mixtral 8×7B | Mixtral 8×22B | Deepseek-MoE-16B | Qwen2-57B-14A |
|---|---|---|---|---|
| NAEE | 1.31h | 1.57h | $\approx 94000$d | $\approx 113000$d |
| **DiEP(Ours)** | **0.23h** | **0.31h** | **0.28h** | **0.34h** |

Table 4: Inference cost analysis on Mixtral 8×7B after expert pruning.

| $r$ | Pruning | Skipping | Avg. Acc | Speedup ↑ | GPU ↓ |
|---|---|---|---|---|---|
| 0% | | | 65.1 | 1.00× | 1.00 × |
| 0% | | ✓ | 64.1 | 1.07 × | 1.00 × |
| 25% | ✓ | | 63.8 | 1.18× | 0.76 × |
| 25% | ✓ | ✓ | 63.3 | 1.21× | 0.76 × |
| 50% | ✓ | | 59.9 | 1.26× | 0.52 × |
| 50% | ✓ | ✓ | 59.6 | 1.28 × | 0.52 × |

memory. Our DiEP enhances inference efficiency via an online expert skipping, which adjusts router weights according to expert similarity with negligible loss in performance. Using half the experts, DiEP retains nearly 92% performance on Mixtral 8×7B, achieving 1.28× token generation speedup and 48% memory savings. We provide more experimental analysis for ablation study in Appendix A.

## 6 Conclusion

In this paper, we propose DiEP, a novel differentiable expert pruning framework that reframes expert selection as a continuous optimization problem. By enabling gradient-based optimization and introducing an adaptive expert skipping mechanism, our DiEP significantly reduces memory usage and accelerates inference while maintaining high model performance. Extensive experiments show that our DiEP outperforms other MoE pruning methods across various language tasks, and sets a new benchmark for efficient Sparse MoE deployment.

# 7 Acknowledge

This research was supported by fundings from the Hong Kong RGC General Research Fund (152169/22E, 152228/23E, 162161/24E, 162116/25E), Research Impact Fund (No. R5060-19, No. R5011-23), Collaborative Research Fund (No. C1042-23GF), NSFC/RGC Collaborative Research Scheme (Grant No. 62461160332 & CRS_HKUST602/24), Areas of Excellence Scheme (AoE/E-601/22-R), and the InnoHK (HKGAI).

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

# A Experimental Appendices

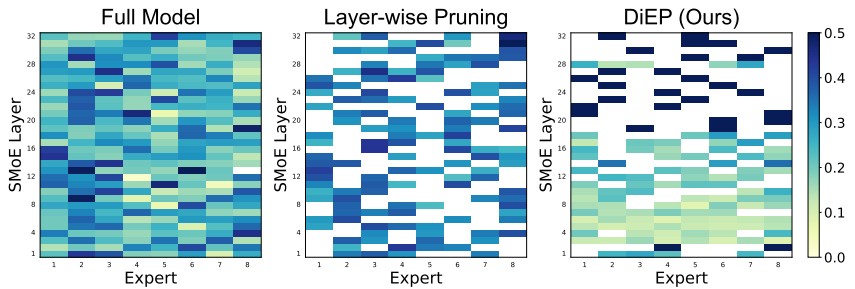

Figure 5: Distribution of expert activation frequencies in the Mixtral 8×7B. The full model (left) uses all experts across all 32 layers, resulting in substantial memory consumption. Layer-wise pruning (middle) enforces uniform expert sparsity per layer. Our DiEP (right) provides a more flexible approach, performing cross-layer expert pruning based on their global contributions.

## A.1 Visualized Analysis of Expert Activation Frequency

To demonstrate the efficacy of adaptive expert pruning in our DiEP, we conducted a comparative analysis of different methods in terms of expert activation frequency. As shown in Figure 5, while the full MoE utilizes almost all experts, there are significant disparities in activation frequencies among different experts, leading to substantial resource waste. Previous methods (i.e., Layer-wise Pruning) apply uniform expert pruning ratios across all layers, overlooking the intra-layer and inter-layer variations and dependencies among experts in different MoE layers. In contrast, our method obtains non-uniform and adaptive expert pruning that varies pruning ratios according to expert-specific characteristics. On Mixtral 8×7B, we observed an increasing trend in expert pruning rates from shallow to higher layers. We attribute this phenomenon the fact that shallow layers primarily process diverse low-level linguistic features, such as part-of-speech tagging and local word ordering, necessitating a larger number of experts to capture detailed linguistic information. Meanwhile, higher layers primarily handle global contextual and semantic information, abstract away from fine-grained details, and thus can operate effectively with fewer experts.

## A.2 Efficiency Analysis for Inference on Deepseek-MoE-16B.

We further verify the efficiency of our adaptive skipping strategy on Deepseek-MoE-16B in Table 5, and it can be observed that our method maintains more than 95% performance of the full model while reducing model size and improving inference efficiency.

Table 5: Inference cost analysis on Deepseek-MoE-16B.

| Model | $r$ | Pruning | Skipping | Avg. Acc | Speedup ↑ | GPU ↓ |
|---|---|---|---|---|---|---|
| Deepseek-MoE-16B | 0% | | | 56.2 | 1.00× | 1.00 × |
| | 0% | | ✓ | 55.7 | 1.04× | 1.00 × |
| | 6.25% | ✓ | | 54.8 | 1.07× | 0.95× |
| | 6.25% | ✓ | ✓ | 54.4 | 1.08× | 0.95× |
| | 12.5% | ✓ | | 54.1 | 1.11× | 0.89× |
| | 12.5% | ✓ | ✓ | 53.6 | 1.13× | 0.89× |

## A.3 Efficiency Analysis for GPU Memory Pruning Cost

To investigate the efficiency of DiEP concerning its memory footprint during the pruning process, we conducted a detailed comparison of GPU memory costs in Table 6, highlighting DiEP's advantages in both efficiency and scalability. ***Computational Efficiency (Time Cost):*** Compared to NAEE (1.31 hours), DiEP's pruning process is 5.7 times faster, requiring only 0.23 hours. DiEP also demonstrates 25.8% faster execution (0.23 hours) compared to MC-MoE (0.31 hours). This indicates a significant

advantage for DiEP in terms of the time required for pruning. ***Memory Optimization (Peak Memory):*** DiEP utilizes 60% less peak memory (139.0GB) than MC-MoE (348.4GB), showcasing superior memory efficiency. While DiEP's peak memory (139.0GB) is 46% higher than NAEE's (95.1GB), this is offset by its dramatically faster pruning time, a factor reflected in its overall resource efficiency. ***Overall Resource Efficiency (Memory-Hour Cost):*** DiEP's memory-hour cost (31.97 GB·h) is 70% lower than that of MC-MoE (108.00 GB·h). Furthermore, DiEP's memory-hour cost is 74% lower than that of NAEE (124.58 GB·h). These results clearly demonstrate that DiEP maintains a lightweight resource footprint while drastically reducing runtime, positioning it as a more resource-efficient choice for MoE pruning.

Table 6: GPU memory pruning cost on Mixtral $8\times7$B.

| Method | Peak Memory(GB) | Time (h) | Memory-hour Cost (GB·h) |
|---|---|---|---|
| NAEE | 95.1 | 1.31 | 124.58 |
| MC-MoE | 348.4 | 0.31 | 108.00 |
| **DiEP (Ours)** | 139.0 | 0.23 | 31.97 |

## A.4 More calibration data validation on adaptability

To further validate DiEP's adaptability, we evaluated its performance on the domain-specific GSM8K dataset using two distinct calibration datasets: the general-purpose C4 dataset and the domain-relevant Math dataset, comparing DiEP against the NAEE method. As detailed in Table 7, the experimental results systematically demonstrate DiEP's advantages across these varied calibration settings. Specifically, when employing the general-purpose C4 calibration data, DiEP achieved consistent improvements over NAEE, outperforming it by +3.93 points at a 50% pruning rate and by +4.96 points at a 25% pruning rate, indicating robust performance gains with common calibration data. Furthermore, when utilizing the domain-specific MATH calibration data, DiEP maintained its superior performance, securing a +1.10 point advantage at 50% pruning and extending this lead to +2.21 points at 25% pruning. These findings collectively underscore DiEP's enhanced generalization capabilities and adaptability across calibration datasets with different data distributions.

Table 7: Adaptability validation on GSM8K using different calibration datasets (C4 and Math).

| Method | Pruning Dataset | r=25% | r=50% |
|---|---|---|---|
| Random | | 36.39 | 0.68 |
| NAEE | C4 | 41.02 | 24.87 |
| **DiEP (Ours)** | C4 | 45.98 | 28.80 |
| NAEE | MATH | 51.25 | 37.07 |
| **DiEP (Ours)** | MATH | 53.46 | 38.17 |

## A.5 Merging Strategy

Inspired by S-SMoE [49], we introduce a merging strategy for DiEP to consolidate redundant experts while preserving their diversity. Specifically, pruned experts are grouped with their most similar retained counterparts based on normalized CKA similarity, which is then normalized by the softmax function as the merging weight. Table 8 demonstrates that the merging strategy further enhances performance under 25% and 50% expert sparsity, which highlights the strong scalability of our DiEP. It not only effectively maintains the performance of the full model but also further restores the diversity of pruned experts by incorporating other orthogonal strategies.

Table 8: Performance analysis when integrating merging strategy.

| Samples | Strategy | MMLU | BoolQ | OpenBookQA | RTE | Avg. |
|---|---|---|---|---|---|---|
| 25% | DiEP | 64.9 | 86.6 | 33.1 | 70.7 | 63.8 |
| | DiEP+Merging | **66.6** | **86.1** | **34.1** | **71.0** | **64.4** |
| 50% | DiEP | 57.9 | 84.0 | 29.6 | 68.2 | 59.9 |
| | DiEP+Merging | **58.2** | **84.0** | **29.8** | **68.8** | **60.2** |

## A.6 Impact of Calibration Data Size

To analyze the impact of calibration data size, we randomly sampled 32, 64, 128, 256, 512, and 1024 sequences from C4 dataset [37] to learn DiEP's intra-layer scores ($\alpha$) and inter-layer scores $\beta$. As shown in Table 10, 128 sequences achieve optimal performance when pruning Mixtral 8×7B from 8 to 6 experts. More importantly, DiEP avoids performance collapse with only 32 samples. We attribute it to KD regularization enforcing DiEP's features aligned with the full model.

Table 9: Performances of expert pruning when changing the number of samples in the calibration dataset.

| Samples | MMLU | BoolQ | OpenBookQA | RTE | Avg. |
|---|---|---|---|---|---|
| 32 | 62.8 | 84.3 | 31.6 | 65.5 | 61.1 |
| 64 | 63.6 | 85.3 | 32.2 | 66.4 | 61.9 |
| 128 | 64.9 | 86.6 | 33.1 | 70.7 | 63.8 |
| 256 | 64.7 | 85.9 | 32.6 | 70.4 | 63.4 |
| 512 | 64.3 | 84.5 | 32.6 | 67.5 | 62.3 |
| 1,024 | 63.7 | 83.9 | 32.8 | 66.3 | 61.9 |

## A.7 Impact of Intra-layer $\alpha$ and Inter-layer $\beta$ Update Ratio

The update ratio is an empirical choice aimed at achieving optimization stability. The high-dimensional $\alpha$ scores learn fine-grained expert rankings, while the low-dimensional $\beta$ scores make coarse-grained, systemic adjustments. Our hypothesis is that allowing $\alpha$ to update more frequently helps stabilize the local expert rankings before the more impactful $\beta$ scores are adjusted.

To rigorously justify our choice, we conducted a new ablation study on the $\alpha$:$\beta$ update ratio. The results, shown below, are for 25% and 50% pruning ratios (formatted as 25%/50%). The experimental results clearly validate our design choice. The 3:1 update ratio consistently achieves the best or near-best performance across almost all tasks and pruning ratios, culminating in the highest average scores for both 25% (63.8) and 50% (59.9) pruning. The data shows that giving more updates to the intra-layer scores ($\alpha$) generally leads to better performance (e.g., 2:1 and 4:1 outperform 1:1 and 1:2). This supports our hypothesis that stabilizing the local expert rankings is crucial. The 3:1 ratio strikes an optimal balance, outperforming both less frequent (e.g., 2:1) and more frequent (e.g., 4:1) $\alpha$ update schedules in terms of average performance. In conclusion, our $\alpha/\beta$ decomposition provides a principled way to model multi-scale redundancy, and our chosen 3:1 update ratio is not arbitrary but is empirically validated to be the most effective schedule for stable and high-performing optimization. We will incorporate these new results into our revised manuscript.

Table 10: Ablation study on the $\alpha$:$\beta$ update ratio.

| $\alpha : \beta$ | MMLU | BoolQ | OpenBookQA | RTE | Avg. |
|---|---|---|---|---|---|
| 1:1 | 65.1/54.3 | 85.2/81.8 | 31.8/29.8 | 67.2/65.4 | 62.3/57.7 |
| 1:2 | 64.5/54.8 | 85.5/79.5 | 31.8/27.6 | 69.0/64.5 | 62.7/56.6 |
| 2:1 | 66.2/56.5 | 85.1/82.5 | 32.4/28.9 | 71.2/67.2 | 63.7/58.7 |
| 2:2 | 65.3/55.1 | 85.6/83.9 | 32.0/27.8 | 67.9/65.1 | 62.7/57.9 |
| 3:1 | 64.9/57.9 | 86.6/84.0 | 33.1/29.6 | 70.7/68.2 | 63.8/59.9 |
| 4:1 | 65.2/56.7 | 85.3/83.7 | 31.4/29.4 | 70.8/66.1 | 63.1/58.9 |

## A.8 Complete Visualized Analysis of Expert Similarity

To validate our motivation regarding the necessity of cross-layer pruning, we first visualized the intra-layer expert similarities in each layer using the CKA similarity metric [22] for Mixtral 8×7B in Figure 6. The analysis reveals significant variations in expert similarities, particularly pronounced in layer 31. Moreover, substantial differences in expert similarities exist between different layers, with layers 28-29 showing higher similarity compared to layers 8-10. Furthermore, we investigate expert-pairs similarities in adjacent layers in Figure 7, which demonstrates varying degrees of expert relationships across layers, exemplified by the strong correlation between expert 6 in layer 30 and expert 5 in layer 31. These cross-layer expert dependencies have been overlooked by previous

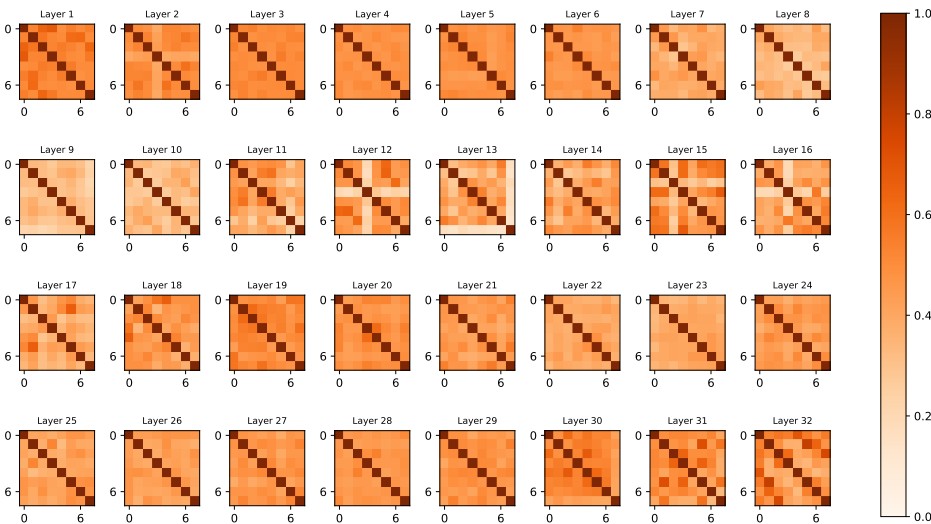

Figure 6: Visualization for feature similarity of expert-pairs within each MoE layer.

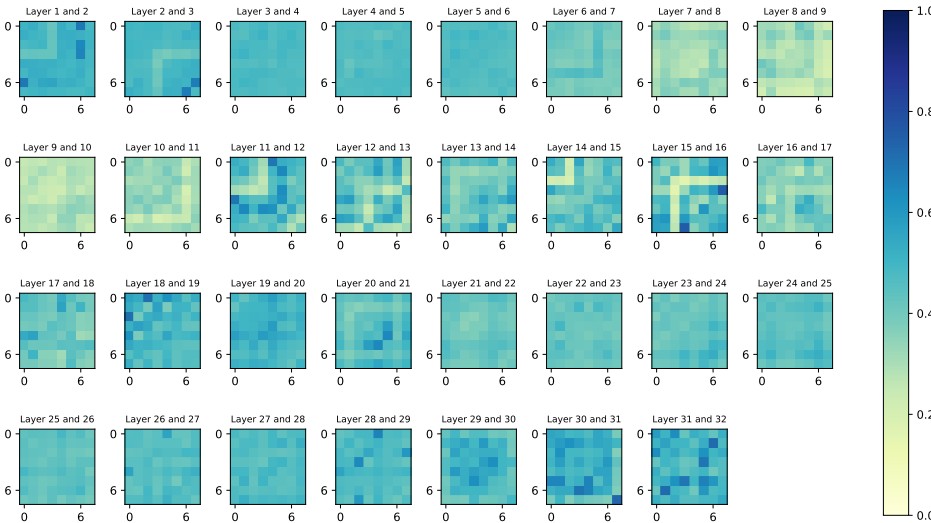

Figure 7: Visualization for feature similarity of expert-pairs across adjacent MoE layers.

pruning methods. Our approach effectively captures both inter-layer and intra-layer variations through alternating differentiable optimization of expert weight $\alpha$ and layer weight $\beta$. In addition, we observed that the learned intra-layer and inter-layer scores do not fully correspond to the visualized inter-layer similarity between expert pairs. It is plausible because we only provide expert similarity across adjacent layers for visualized analysis. However, our DiEP can learn expert redundancy and dependency across all MoE layers.

## A.9 Hyperparameters Analysis

There are two key hyperparameters, including the number of epochs during differentiable search and the value of the weight $\lambda$ between reconstruction regularization term and cross-entropy loss of the overall objective function in Eq. 7. We first analyze the impact of $\lambda$ by varying its value in $\{5, 10, 15, 20, 30\}$. Figure 8a demonstrates that optimal performance is achieved with $\lambda = 0.01$ for both 25% and 50% expert sparsity. Additionally, we investigate how the number of epochs affects

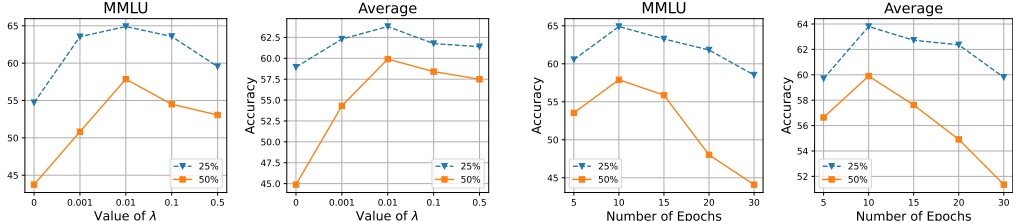

(a) Hyperparameters analysis in terms of value of weight coefficient $\lambda$ in Eq. 7.

(b) Hyperparameters analysis in terms of the number of epochs.

Figure 8: Hyperparameters analysis in terms of the number of clients and weight coefficient $\lambda$ on Mixtral8×7B under 25% and 50% expert sparsity.

model performance. As shown in Figure 8b, DiEP achieves optimal results when trained for 10 epochs under both 25% and 50% expert sparsity settings.

### A.10 Results on More Datasets.

We provide more experimental results on more datasets including ARC-c, ARC-e[9], HellaSwag [47] and WinoGrand [38] on Mixtral 8×7B, and our DiEP is much better than NAEE across all tasks. As shown in Figure 11, these results further demonstrate the effectiveness of our proposed method.

Table 11: Zero-shot evaluation result on more datasets, including ARC-c, ARC-e, HellaSwag, WinoGrande.

| Model | Method | ARC-c | ARC-e | HellaSwag | WinoGrande |
|---|---|---|---|---|---|
| Mixtral 8×7B | NAEE | 51.62 /48.89 | 81.94/78.16 | 61.60/57.66 | 75.37/72.85 |
| | DiEP(Ours) | **52.54/49.26** | **83.31/82.52** | **63.22/58.96** | **76.03/73.55** |

## B    Theoretical Appendices

### B.1    Algorithm Pipeline of DiEP

---

**Algorithm 1: DiEP** – Differentiable Expert Pruning

---

**Input:** Model inputs $\boldsymbol{x}$, targets $\boldsymbol{y}$, initial intra-layer scores $\alpha^0$, initial inter-layer scores $\beta^0$, regularization coefficient $\lambda$.

**while** *not converged* **do**

$\quad$ /* Update intra-layer scores (fix $\beta$) $\qquad\qquad\qquad\qquad\qquad\qquad\qquad\qquad\qquad$ */

$\quad$ Update $\alpha^{t+1}$ by descending:

$$\nabla_\alpha\big(\mathcal{L}_{ce}(\boldsymbol{y}, \mathcal{F}'(\boldsymbol{x}; \alpha^t, \beta^t)) + \lambda\Phi(\alpha^t, \beta^t)\big)$$

$\quad$ /* Update inter-layer scores (fix $\alpha$) $\qquad\qquad\qquad\qquad\qquad\qquad\qquad\qquad\qquad$ */

$\quad$ Update $\beta^{t+1}$ by descending:

$$\nabla_\beta\big(\mathcal{L}_{ce}(\boldsymbol{y}, \mathcal{F}'(\boldsymbol{x}; \alpha^{t+1}, \beta^t)) + \lambda\Phi(\alpha^{t+1}, \beta^t)\big)$$

$\quad$ Set $t \leftarrow t + 1$

**Output:** Optimized intra-layer importance scores $\alpha$ and inter-layer importance scores $\beta$.

---

### B.2    Convergence Analysis of DiEP

Let $\Theta := \{(\alpha, \beta)\}$ be the parameter space, where $\alpha \in \mathbb{R}^{NL}$ and $\beta \in \mathbb{R}^L$. Denote $\theta_1 = \alpha$ and $\theta_2 = \beta$. The overall objective of **DiEP** is

$$\mathcal{L}(\theta_1, \theta_2) \;=\; \mathcal{L}_{ce}\big(\mathbf{y}, \mathcal{F}'(\mathbf{x}; \theta_1, \theta_2)\big) \;+\; \lambda\big\|\mathcal{F}'(\mathbf{x}; \theta_1, \theta_2) - \mathcal{F}(\mathbf{x})\big\|_F.$$

**Assumptions.**

*A1*. *Lower-Boundedness.* $\inf\limits_{(\alpha,\beta)\in\Theta} \mathcal{L}(\alpha,\beta) > -\infty$.

*A2*. *Lipschitz Smoothness.* $\nabla_{\theta_i}\mathcal{L}$ is $L_i$-Lipschitz continuous, i.e. $\|\nabla_{\theta_i}\mathcal{L}(u) - \nabla_{\theta_i}\mathcal{L}(v)\| \le L_i\|u - v\|$ for $i \in \{1,2\}$.

*A3*. *Stepsizes.* Fixed learning rates satisfy $0 < \eta_i < \frac{2}{L_i}$ for $i \in \{1,2\}$.

*A4*. *Level-Set Boundedness.* The set $\{(\alpha,\beta) \in \Theta : \mathcal{L}(\alpha,\beta) \le \mathcal{L}(\alpha^0,\beta^0)\}$ is compact.

**Algorithmic update.** For $t = 0, 1, \ldots$

$$\theta_i^{t+1} = \theta_i^t - \eta_i\,\nabla_{\theta_i}\mathcal{L}(\theta_1^t, \theta_2^t), \quad i \in \{1,2\}. \tag{13}$$

**Lemma 1** (Descent). *Under A2–A3,* $\mathcal{L}(\theta_1^{t+1}, \theta_2^{t+1}) \le \mathcal{L}(\theta_1^t, \theta_2^t) - \sum_{i=1}^{2} \frac{(2-\eta_i L_i)}{2\eta_i}\big\|\theta_i^{t+1} - \theta_i^t\big\|^2$.

*Proof.* Apply the standard descent lemma to each block update in (13). $\qquad\square$

**Corollary 1** (Monotonicity and Bounded Iterates). *Assumptions A1–A4 imply* $\big\{\mathcal{L}(\theta_1^t, \theta_2^t)\big\}_{t\ge 0}$ *is monotonically non-increasing and convergent, and* $\{(\theta_1^t, \theta_2^t)\}_{t\ge 0}$ *is bounded.*

**Lemma 2** (Vanishing Updates). $\lim\limits_{t\to\infty}\big\|\theta_i^{t+1} - \theta_i^t\big\| = 0,\ i \in \{1,2\}$.

*Proof.* Summing the non-negative terms in Lemma 1 over $t$ gives a telescoping series dominated by $\mathcal{L}(\theta_1^0, \theta_2^0) - \inf\mathcal{L}$; hence the series of squared update norms is finite. $\qquad\square$

**Theorem 1** (Subsequence Convergence to Critical Points). *Under A1–A4, the sequence* $\{(\theta_1^t, \theta_2^t)\}_{t\ge 0}$ *generated by* (13) *possesses at least one convergent subsequence, and* every *limit point* $(\theta_1^\star, \theta_2^\star)$ *satisfies* $\nabla_{\theta_1}\mathcal{L}(\theta_1^\star, \theta_2^\star) = \mathbf{0}$ *and* $\nabla_{\theta_2}\mathcal{L}(\theta_1^\star, \theta_2^\star) = \mathbf{0}$; *i.e. it is a critical point of* $\mathcal{L}$.

*Proof.* By Corollary 1 and **A4**, $\{(\theta_1^t, \theta_2^t)\}$ lies in a compact set; hence the Bolzano–Weierstrass theorem guarantees a convergent subsequence $(\theta_1^{t_k}, \theta_2^{t_k}) \to (\theta_1^\star, \theta_2^\star)$. Lemma 2 gives $\theta_i^{t_k+1} - \theta_i^{t_k} \to \mathbf{0}$. Dividing (13) by $\eta_i$ and taking $k\to\infty$ yields $\nabla_{\theta_i}\mathcal{L}(\theta_1^\star, \theta_2^\star) = \mathbf{0}$ for $i = 1, 2$. $\qquad\square$

**Discussion.** Theorem 1 aligns with the classical results of block coordinate descent [42] while instantiating them for our two-block differentiable pruning objective. It guarantees that DIEP converges (up to subsequences) to first-order stationary points under standard smoothness and stepsize conditions.

## C   Limitations

While our proposed DiEP method achieves strong performance in MoE pruning, several limitations remain. Due to computational constraints, our main experiments cannot be conducted on some larger-scale MoE models, like Deepseek V3 [27], and Qwen2.5-Max [44]. Furthermore, our study primarily focuses on language models, leaving the effectiveness of DiEP in multimodal MoE architectures unexplored. Investigating whether our approach can achieve competitive performance on Vision-Language tasks, such as MoE-LLaVA [26], remains an important direction for future research.

## D   Broader impacts

The development of DiEP, a method for compressing large Mixture-of-Experts (MoE) models, carries several potential societal impacts, both positive and negative. Positive impacts: In regions or scenarios where access to high-performance computing infrastructure is limited, DiEP can enable the deployment of capable AI models that would otherwise be infeasible. This could support applications in education, healthcare, and public services in underserved communities. Negative impacts: As AI models become more capable and efficient, they may automate tasks currently performed by humans, potentially leading to job displacement in certain sectors. While DiEP aims at efficiency, the broader trend of AI advancement contributes to this concern.

