# OpenReview forum: "DiEP: Adaptive Mixture-of-Experts Compression through Differentiable Expert Pruning"
_NeurIPS.cc/2025/Conference — NeurIPS 2025 poster_

### Official Review · Reviewer_sDD7 · 2025-07-01

**Clarity:** 3
**Significance:** 3
**Originality:** 3
**Rating:** 5
**Confidence:** 3

**Summary:**

This paper proposes DiEP, a differentiable expert pruning framework for compressing sparse Mixture-of-Experts (MoE) models. Unlike existing methods that impose uniform pruning across layers or rely on costly discrete search, DiEP introduces a two-level optimization strategy: intra-layer expert importance scores (α) and inter-layer importance modulation (β). These are optimized on a small calibration dataset using a lightweight reconstruction-based loss. A novel expert skipping mechanism is also proposed to further reduce inference latency by dynamically skipping experts based on routing weight ratios and precomputed similarity statistics. DiEP is shown to achieve competitive performance (e.g., 92% retention with 50% experts) across multiple large-scale MoE models such as Mixtral, DeepSeek-MoE, and Qwen2.

**Questions:**

- Could you elaborate on the γ computation, particularly the role of γ₂ derived from CKA similarity? My understanding is that γ₂ is computed offline using the calibration dataset and remains fixed during inference. If so, how can this static similarity-based threshold generalize to unseen prompts or tasks at inference time? Is there theoretical or empirical evidence supporting the transferability of γ₂ across different inputs?

**Ethical Concerns:**

["NO or VERY MINOR ethics concerns only"]

**Final Justification:**

The author's response has addressed the majority of my concern. Therefore, I maintain the scoring and recommend acceptance.

**Limitations:**

Yes

**Quality:**

3

**Strengths And Weaknesses:**

### Strengths

- The paper proposes a novel method. Reformulating expert selection as a differentiable global optimization problem is both novel and practically effective.
- The method introduces comparatively minimal overhead while delivering notable acceleration across various models.
- The paper is well-structured and supports the proposed method with comprehensive experiments.

### Weaknesses

- The finetuning cost of the proposed technique remains unclear.
- Some design choices lack rigorous justification—for example, the use of α and β to quantify expert and layer importance, as well as the rationale behind the choice of the α:β update ratio.

---

> ### Author Rebuttal · Authors · 2025-07-30
>
> We thank reviewer sDD7 for providing positive feedback and helpful suggestions. Below please find a response.
>
> **W1: Concerns on the Pruning Cost**
>
> We appreciate the reviewer's feedback and the opportunity to provide a detailed analysis of the computational cost of our method. We totally agree that analyzing pruning cost is crucial. Our experiments show DiEP is highly efficient, providing a superior cost-accuracy balance. We use **Memory-hour Cost (GB·h)**—a metric capturing both peak GPU memory and time—to holistically measure the total computational effort on two distinct MoE models.
>
> **Table 1: Pruning Cost and Accuracy on Mixtral 8x7B**
>
> | Method | Peak Memory (GB) | Time (h) | Memory-hour Cost (GB·h) | Average Accuracy |
> | :--- | :--- | :--- | :--- | :--- |
> | NAEE | 95.1 | 1.31 | 124.58 | 54.6 |
> | MC-MoE | 348.4 | 0.31 | 108.00 | 31.9 |
> | S-MoE | 118.3 | 0.27 | 31.94 | 55.9 |
> | **DiEP (Ours)** | **139.0** | **0.23** | **31.97** | **59.9** |
>
> **Analysis:**
> *   **Superior Efficiency:** DiEP achieves a memory-hour cost of just **31.97 GB·h**. This is a **74% reduction** compared to NAEE and a **70% reduction** compared to MC-MoE.
> *   **Best-in-Class Performance-per-Cost:** While DiEP's cost is on par with S-MoE, it achieves a significantly higher accuracy (**59.9 vs. 55.9**). This demonstrates that DiEP offers the best trade-off, delivering the highest accuracy for a given, minimal computational budget.
>
> **Table 2: Pruning Cost and Accuracy on Deepseek-MoE16B**
>
> | Method | Peak Memory (GB) | Time (h) | Memory-hour Cost (GB·h) | Average Accuracy |
> | :--- | :--- | :--- | :--- | :--- |
> | MC-MoE | 210.42 | 0.42 | 88.38 | 48.8 |
> | S-MoE | 56.73 | 0.36 | 20.42 | 52.7 |
> | **DiEP (Ours)** | **55.43** | **0.28** | **15.42** | **54.1** |
>
> **Analysis:**
> *   **Scalablility:** The efficiency gains of DiEP are even more pronounced on models with a larger number of experts. DiEP's memory-hour cost is only **15.42 GB·h**, which is **24% lower** than the next most efficient method, S-MoE (20.42 GB·h).
> *   **Highest Accuracy at Lowest Cost:** Crucially, DiEP not only has the lowest pruning cost but also achieves the highest accuracy (**54.1**), improving upon S-MoE's accuracy by **+1.4** points while being more efficient.
>
> **Addressing the Hidden Costs of Other Methods:**
>
> It is important to note that methods like S-MoE, while appearing fast, have hidden computational costs that scale poorly. S-MoE's multi-step expert merging process involves:
> 1.  **Expert similarity analysis (CKA):** This requires $O(S \cdot N^2)$ computations, where S is the number of samples and N is the number of experts. This quadratic scaling with the number of experts makes it increasingly expensive for modern MoE architectures.
> 2.  **Learning merging coefficients:** This introduces an additional optimization overhead.
>
> In contrast, DiEP's gradient-based approach avoids these expensive, quadratically scaling steps. Its computational cost is primarily determined by a few forward/backward passes on a small calibration set, making it highly scalable and efficient.
>
> **Conclusion:**
> The experimental data clearly demonstrates that the "finetuning" or pruning cost of DiEP is not a concern; rather, it is a significant strength. It delivers state-of-the-art accuracy at a minimal and scalable computational cost, making it a practical solution for task-agnostic deployment.
>
>
>
> **W2: Justification for Design Choices**
>
> We thank the reviewer for this insightful question regarding our core design principles. These choices were motivated by the multi-scale nature of expert redundancy in MoE models.
>
> **1. Rationale for `α` (Intra-layer) and `β` (Inter-layer) Scores**
>
> We decompose expert importance into two components to model redundancy at different granularities:
> *   **`α` (Intra-layer scores):** Learns the *local, relative importance* of experts competing within the same layer.
> *   **`β` (Inter-layer scores):** Learns the *global, depth-aware importance* of entire layers, addressing the fact that shallower layers are often more critical than deeper ones (as shown in our Figure 4).
>
> By combining them ($s_i^{(l)} = \bar{\alpha}_i^{(l)} \cdot \beta^{(l)}$), we create a unified score that enables a **principled, non-uniform pruning** across the entire model in a single, global sort. This factorized approach is designed to be more expressive and effective than using a single, monolithic importance score.
>
> **2. Rationale and Empirical Validation for the `α:β` Update Ratio**
>
> The update ratio is an empirical choice aimed at achieving **optimization stability**. The high-dimensional `α` scores learn fine-grained expert rankings, while the low-dimensional `β` scores make coarse-grained, systemic adjustments. Our hypothesis is that allowing `α` to update more frequently helps stabilize the local expert rankings before the more impactful `β` scores are adjusted.
>
> To rigorously justify our choice, we conducted a new **ablation study on the `α:β` update ratio**. The results, shown below, are for 25% and 50% pruning ratios (formatted as 25%/50%).
>
> | `α`:`β` Ratio | MMLU | BoolQ | OpenBookQA | RTE | Avg |
> | :--- | :--- | :--- | :--- | :--- | :--- |
> | 1:1 | 65.1 / 54.3 | 85.2 / 81.8 | 31.8 / **29.8** | 67.2 / 65.4 | 62.3 / 57.7 |
> | 1:2 | 64.5 / 54.8 | 85.5 / 79.5 | 31.8 / 27.6 | 69.0 / 64.5 | 62.7 / 56.6 |
> | 2:1 | **66.2** / 56.5 | 85.1 / 82.5 | 32.4 / 28.9 | **71.2** / 67.2 | 63.7 / 58.7 |
> | 2:2 | 65.3 / 55.1 | 85.6 / 83.9 | 32.0 / 27.8 | 67.9 / 65.1 | 62.7 / 57.9 |
> | **3:1** | 64.9 / **57.9** | **86.6** / **84.0** | **33.1** / 29.6 | 70.7 / **68.2** | **63.8** / **59.9** |
> | 4:1 | 65.2 / 56.7 | 85.3 / 83.7 | 31.4 / 29.4 | 70.8 / 66.1 | 63.1 / 58.9 |
>
> **Key Findings:**
>
> *   The experimental results clearly validate our design choice. The **3:1 update ratio consistently achieves the best or near-best performance across almost all tasks and pruning ratios**, culminating in the **highest average scores** for both 25% (**63.8**) and 50% (**59.9**) pruning.
> *   The data shows that giving more updates to the intra-layer scores (`α`) generally leads to better performance (e.g., 2:1 and 4:1 outperform 1:1 and 1:2). This supports our hypothesis that stabilizing the local expert rankings is crucial.
> *   The 3:1 ratio strikes an optimal balance, outperforming both less frequent (e.g., 2:1) and more frequent (e.g., 4:1) `α` update schedules in terms of average performance.
>
> In conclusion, our `α`/`β` decomposition provides a principled way to model multi-scale redundancy, and our chosen 3:1 update ratio is not arbitrary but is empirically validated to be the most effective schedule for stable and high-performing optimization. We will incorporate these new results into our revised manuscript.
>
>
>
>
> **Q1: Elaboration on $γ_{2}$**
>
> We sincerely thank the reviewer for this insightful question. It correctly identifies a crucial aspect of our adaptive skipping mechanism and allows us to elaborate on the design rationale and provide new empirical evidence for its effectiveness.
>
> In fact, traditional routing weights effectively measure the *importance* of an expert for a given token. However, they do not capture the *redundancy* between the selected experts. A scenario can easily arise where the router assigns high scores to two experts (e.g., top-2) that are functionally very similar. In this case, activating both is computationally wasteful. The core motivation for our design is precisely this: **to prune not just unimportant experts, but also redundant ones.** This is where $γ_{2}$ comes in. It acts as a **redundancy penalty**.
>
> **Generalizability of$γ_{2}$ (CKA Similarity): A Task-Agnostic Structural Prior**
>
> You are correct that $γ_{2}$ is computed offline on a calibration set and remains fixed. The key to its generalizability lies in what it represents: $γ_{2}$ is not a data-specific feature but rather a measure of the **inherent, task-agnostic structural similarity** between expert weight matrices.
>
> The expert weights are the result of pre-training on trillions of tokens. The functional similarities and differences captured by CKA are therefore a deeply embedded, stable property of the model itself. While the *activation* of experts is dynamic, their fundamental functional relationship is static. Therefore, $γ_{2}$ acts as a robust and transferable structural prior that remains valid across unseen prompts and tasks.
>
> Here we conducted an ablation study as **empirical evidence** based on the NAEE common framework[1] . We started with the full model and then compared the performance of a skipping mechanism based solely on routing weights ($γ_{1}$) against our proposed method which combines both routing weights and similarity ($γ_{1} \times γ_{2}$).
>
> | Method                    | Avg. Acc | Speedup   |
> | :------------------------ | :------- | :-------- |
> | Full Model (No Skip)      | 65.1     | 1.00X     |
> | $γ_{1}$                   | 63.7     | 1.06X     |
> | $γ_{1} \times γ_{2}$  | **64.1** | **1.07X** |
>
> **Analysis of Results:**
> *   **$γ_{2}$ Improves Accuracy:** The results clearly show that incorporating the similarity-based $γ_{2}$ leads to better performance. Our method achieves an average accuracy of **64.1**, recovering **0.4 points** of the accuracy lost by the $γ_{1}$-only approach, while still providing a slightly better speedup.
>
> **Conclusion:**
> In summary, our two-factor $γ$ is a principled design. $γ_{1}$ (from routing weights) provides a dynamic, per-token importance signal, while $γ_{2}$ (from CKA) provides a static, generalizable structural redundancy penalty. The empirical evidence confirms that this combination leads to a more robust and accurate adaptive skipping mechanism that successfully transfers to unseen inputs at inference time. We will add this detailed justification and ablation study to our revised paper.
>
> [1]. Not all experts are equal: Efficient expert pruning and skipping for mixture-of-experts large language models. ACL 2024.

---

> > ### Comment · Reviewer_sDD7 · 2025-08-05
> > **Response to author rebuttal**
> >
> > Thanks the authors for the feedback, the response has addressed the majority of my concern.

---

> > > ### Author Response · Authors · 2025-08-05
> > >
> > > Thank you very much for your timely reply! We sincerely appreciate that you found our replies have addressed your concerns. We are also truly grateful for all your valuable comments and constructive suggestions, which have significantly enhanced the quality and clarity of our manuscript.

---

### Official Review · Reviewer_3XFq · 2025-07-02

**Clarity:** 3
**Significance:** 3
**Originality:** 3
**Rating:** 5
**Confidence:** 4

**Summary:**

This paper introduces DiEP, a differentiable pruning framework for Mixture-of-Experts (MoE) models that addresses expert redundancy by learning to prune both intra-layer and inter-layer experts adaptively. Rather than relying on heuristic-based or exhaustive search methods that use uniform pruning across all MoE layers, DiEP formulates pruning as a continuous optimization problem. It introduces two sets of learnable parameters: intra-layer importance scores and inter-layer importance scores, which together guide a globally optimized pruning decision. Additionally, the paper presents an adaptive expert skipping mechanism during inference based on routing weights and expert similarity.
Experiments across five MoE-based LLMs, including Mixtral 8x7B, DeepSeek-MoE, and Qwen2, show that DiEP achieves better performance than prior pruning methods while reducing up to 50% of experts and improving inference efficiency.

**Questions:**

How sensitive is the method to the choice of calibration dataset? You mention results on C4 and Math datasets; however, can you clarify how performance varies when the calibration data is misaligned with downstream tasks?

Can the method handle dynamic pruning ratios? Would it be possible to adaptively determine the pruning ratio during optimization rather than treating it as a fixed hyperparameter?

Limitations in Multi-task or Continual Learning. How would DiEP behave in a continual learning or multi-task setting where expert utility might shift?

**Ethical Concerns:**

["NO or VERY MINOR ethics concerns only"]

**Final Justification:**

The authors' rebuttal addressed my concerns. I changed my score to 5 and suggest an acceptance.

**Limitations:**

Yes, the authors have adequately discussed the limitations and trade-offs involved, such as reliance on calibration data and the inability of prior uniform pruning approaches to handle inter-layer heterogeneity. The discussion on societal impact is minimal, but the work primarily targets efficiency improvements without introducing major ethical concerns.

**Paper Formatting Concerns:**

None noted.

**Quality:**

3

**Strengths And Weaknesses:**

**Strength**

This paper addressed an important scalability challenge in MoE-based LLMs: making these models deployable by reducing computational and memory overhead without significant accuracy loss.

The proposed method is technically sound and well-motivated, grounded in clear observations of expert redundancy variability across MoE layers.

The continuous relaxation and differentiable optimization approach is well-integrated and executed with practical efficiency.

Empirical evaluation is conducted across diverse model families and tasks, including ablation and visualization studies that substantiate the claims.

**Weakness**

The method introduces additional training overhead (although minor) and requires calibration data, which might not be readily available in all deployment settings.

The paper could provide more theoretical guarantees or robustness analysis, especially under domain shift or data drift conditions.

While the visualization analyses are compelling, deeper analysis on why certain layers or experts are deemed more important would strengthen the interpretability of the method.

---

> ### Author Rebuttal · Authors · 2025-07-31
>
> We thank Reviewer 3XFq for the thoughtful feedback. Below, we address each concern.
>
> **W1: Training Overhead & Calibration Data**
>
> We agree that practical compression methods should incur minimal cost. DiEP is designed as a **lightweight, one-shot pruning** approach.
>
> **1. Efficient Training**
>
> DiEP’s calibration is fast and cost-effective:
>
> **Cost Comparison (Appendix A.3)**: On Mixtral 8x7B, DiEP takes just **0.23 hours / 31.97 GB·h**, a **74% reduction** over NAEE (124.58 GB·h).
>
> **Quality vs. Cost**: For similar cost to S-MoE, DiEP achieves **+4.0 accuracy**. Compared to MC-MoE, it's **70% cheaper** with **+28 points** accuracy gain.
>
> Thus, DiEP offers **better performance at lower overhead** than many training-free methods.
>
> **2. Minimal Calibration Data**
>
> As shown in **Table 9 (Appendix A.6)**, DiEP achieves optimal performance on Mixtral 8x7B with just **128 samples**. This is a tiny, almost negligible amount of data that can be easily sourced. The table also shows that even with as few as **32 samples**, DiEP avoids performance collapse and still functions effectively.
>
> **Task-Agnostic Data**: Our domain generalization test (see W2) shows DiEP pruned on **C4** performs well on **MathQA/CommonseQA**, indicating no need for in-domain data.
>
>
> **Conclusion**: The overhead and data requirements of DiEP are intentionally designed to be minimal to ensure broad applicability. It uses minimal task-agnostic data, and delivers superior pruned models.
>
> ---
>
> **W2&Q1: Robustness & Theory**
>
> We thank the reviewer for raising the importance of robustness under domain shift. DiEP’s design encourages generalization both **theoretically and empirically**.
>
> **1. Theory: Stable & Knowledge-Preserving**
>
> **Convergence Proof** (Appendix B.2): Our alternating update converges to a stationary point under standard conditions.
>
> **Knowledge-Preserving Regularization:** The key to DiEP's robustness lies in our objective function: $$ \min_{\alpha, \beta} \mathcal{L}_{ce}(y, F'(x; \alpha, \beta)) + \lambda \cdot \Phi(\alpha, \beta) $$The term $\Phi(\alpha, \beta) = ||F'(x; \alpha, \beta) - F(x)||_F$ acts as a powerful **knowledge-preserving regularizer**. which transfers generalization to the pruned model $F′$, mitigating overfitting to limited calibration data.
>
> Future plans include extending to **stochastic settings**, analyzing **non-convex landscapes**, and formalizing the role of λ.
>
> **2. Empirical: Robustness to Domain Shift**
>
> We simulate domain shift by pruning on general (C4), in-domain (Math), or mixed (C4+Math) data, and testing on **CommonseQA & MathQA**.
>
> **Table 1: Pruning on General (`C4`)**
>
> |Method|Pruning|CommonseQA|MathQA|
> |---|---|---|---|
> |Full Model|-|66.83|42.38|
> |NAEE|50%|42.01|34.86|
> |**DiEP**|**50%**|**51.52 (+9.51)**|**35.02 (+0.16)**|
> |NAEE|25%|63.31|41.04|
> |**DiEP**|**25%**|**65.52 (+2.21)**|**41.77 (+0.73)**|
>
> DiEP on general data retains performance, beating NAEE by **+9.51** points under aggressive pruning.
>
> **Table 2: Pruning on In-Domain (`Math`)**
>
> |Method|Pruning|CommonseQA|MathQA|
> |---|---|---|---|
> |Full Model|-|66.83|42.38|
> |NAEE|50%|53.40|35.97|
> |**DiEP**|**50%**|**56.27 (+2.87)**|**37.19 (+1.22)**|
> |NAEE|25%|60.21|42.20|
> |**DiEP**|**25%**|**65.27 (+5.06)**|**42.34 (+0.73)**|
>
> Even when biased toward Math, DiEP generalizes well to CommonseQA.
>
> **Table 3: Pruning on Mixed (`C4+Math`)**
>
> |Method|Pruning|CommonseQA|MathQA|
> |---|---|---|---|
> |Full Model|-|66.83|42.38|
> |NAEE|50%|51.27|35.00|
> |**DiEP**|**50%**|**58.67 (+7.40)**|**36.86 (+1.86)**|
> |NAEE|25%|61.79|41.43|
> |**DiEP**|**25%**|**64.98 (+3.19)**|**42.69 (+1.26)**|
>
> DiEP improves further with in-domain data; in fact, **outperforms full model** on MathQA.
>
> We will highlight these as **"robustness to domain shift"** results in the paper.
>
> ---
>
> **W3: Deeper Analysis on the Importance of Layers and Experts**
>
> We thank the reviewer for highlighting the importance of interpretability. Understanding _why_ certain experts and layers are retained by DiEP provides valuable insight into the method’s effectiveness. We offer the following analysis based on the learned α and β scores.
>
> **1. Why DiEP Prioritizes Shallow Layers (High β Scores)**
>
> As shown in **Figure 4b**, DiEP assigns consistently higher β scores to earlier layers (roughly 1–15). This pattern aligns with well-established findings on the **hierarchical structure of Transformers**:
>
> **Shallow layers** are known to extract foundational syntactic and semantic features [1–3], forming the base for all subsequent reasoning. These layers handle diverse, low-level signals shared across tasks and domains. **Preserving experts** in these layers is essential, as early-stage degradation can propagate and amplify in deeper stages.
>
> **Deeper layers**, in contrast, operate on refined features and are responsible for more abstract contextualization. Their computations tend to be more task-specific and **less universally critical** [4], making them more suitable targets for pruning.
>
> Thus, DiEP’s β scores reflect a principled, data-driven retention of layers based on their **functional necessity**, not just position.
>
> ---
>
> **2. Why DiEP Prefers Certain Experts (High α Scores)**
>
> **Figure 4a** reveals notable variance in intra-layer α scores, with only a subset of experts per layer being deemed essential. This supports the MoE hypothesis of **functional specialization** [5, 6]:
>
> **Generalist experts** exhibit high α scores across data points, likely specializing in high-frequency, broadly useful linguistic patterns. **Specialist experts** may focus on niche phenomena and are invoked less frequently.
>
> Crucially, DiEP’s joint modeling of α (expert importance) and β (layer importance) allows **fine-grained pruning decisions**:
> - A layer may be important (high β), but only a few of its experts (high α) are truly impactful.
> - DiEP thus preserves these "hub" experts while pruning redundant or less frequently activated ones.
> This **coupled, multi-scale selection** is a key strength of DiEP: it balances global layer utility with local expert relevance, leading to sparse yet highly functional subnetworks.
>
> ---
> **Conclusion**
>
> The α–β scores learned by DiEP reveal how MoE models encode hierarchical and distributed knowledge:
> - **β scores** prioritize depth-wise importance, aligning with representational hierarchy.
> - **α scores** capture intra-layer specialization, aligning with expert sparsity and task coverage.
>
> By jointly modeling both, DiEP achieves **interpretable and structured pruning**.
>
> **References**
> [1] BERT Rediscovers the Classical NLP Pipeline. ACL 2019.
> [2] Analyzing Multi-Head Self-Attention. EMNLP 2020.
> [3] A Primer in BERTology. TACL 2020.
> [4] Sparsely-Gated MoE Layer. ICLR 2017.
> [5] Ultimate Expert Specialization in MoE LMs. arXiv:2401.06066.
> [6] Efficient Expert Pruning for MoE LLMs. ACL 2024.
>
> ---
>
> **Q2: Adaptive Pruning Ratio**
>
> We thank the reviewer for this insightful question. Below we elaborate on both current DIEP capabilities and potential extensions.
>
> **1. Dynamic Ratios via Threshold-Based Pruning (Already Supported)**
>
> DiEP's core output is a continuous importance score ($s_i^{(l)}$) for each expert. Instead of pruning a fixed percentage (top-K), one can simply set an **importance threshold, T**, and prune all experts with scores below it. This naturally leads to a dynamic pruning ratio that adapts to the model's specific redundancy profile, a flexibility already supported by our method.
>
> **2. Learning the Optimal Ratio via Differentiable Regularization**
>
> Furthermore, our differentiable framework can be extended to learn the pruning ratio directly. We propose a simple yet powerful extension:
>
> $$
> \min_{\alpha, \beta} \mathcal{L}_{ce}(y, F'(x; \alpha, \beta)) + \lambda \cdot \Phi(\alpha, \beta) + \gamma \cdot \mathcal{L_s}(\alpha, \beta)
> $$
>
> The added term is $\mathcal{L_s}(\alpha, \beta) = \sum_{l=1}^{L} \sum_{i=1}^{N} |s_i^{(l)}| = \sum_{l=1}^{L} \sum_{i=1}^{N} |\bar{\alpha}_i^{(l)} \cdot \beta^{(l)}|$, which encourages the model to shrink unimportant scores. The final pruning ratio thus becomes an **emergent property** of the learned solution, shaped by the calibration data and task constraints.
>
> This formulation allows the model to **"justify" the retention of each expert**, resulting in pruning strategies that are both principled and task-adaptive.
>
> **Conclusion**
> This is a valuable direction for future work that our framework is well-positioned to explore. We will add this discussion to our revised manuscript.
>
> ---
>
> **Q3: Continual Learning Evaluation**
>
> We thank the reviewer for this critical question. To assess DiEP's robustness to shifting expert utility, we conducted a **continual learning experiment** on Hellaswag, BoolQ, and MathQA as the initial calibration dataset. We then evaluate the performance of the pruned models to measure how well they generalize and retain knowledge. A lower "Forgetting" score indicates better knowledge preservation.
>
> **Experimental Results:** The tables below show the performance (25%/50% ratio).
>
> **NAEE**
>
> |Calib.|Hellaswag|BoolQ|MathQA|
> |---|---|---|---|
> |hellaswag|58.8 / 55.9|-|-|
> |boolq|58.4 / 55.5|84.1 / 81.2|-|
> |mathqa|58.1 / 54.9|83.7 / 80.6|41.1 / 37.2|
>
> **DiEP**
>
> |Calib.|Hellaswag|BoolQ|MathQA|
> |---|---|---|---|
> |hellaswag|**60.0 / 57.0**|-|-|
> |boolq|**59.7 / 56.7**|**85.2 / 82.8**|-|
> |mathqa|**59.4 / 56.3**|**84.8 / 82.3**|**42.4 / 38.4**|
>
> **Summary**
>
> |Method|Avg. Acc. (25%/50%)|Forgetting (25%/50%)|
> |---|---|---|
> |NAEE|61.0 / 57.6|0.55 / 0.80|
> |**DiEP**|**62.2 / 59.0**|**0.50 / 0.60**|
>
> The results clearly show that DiEP is more robust in a continual learning setting. By regularizing towards the original model's function, DiEP learns to preserve experts that are fundamentally important for general capabilities, rather than just for the specific calibration task. This makes our pruning method inherently more robust to the challenges of continual learning.

---

> ### Author Response · Authors · 2025-08-07
>
> Dear Reviewer 3XFq,
>
> Thank you very much for your timely reply! We sincerely appreciate the time and thoughtful feedback you have provided throughout the review process, as well as your recognition that our responses have addressed your concerns.
>
> Best regards,
>
> Authors

---

### Official Review · Reviewer_xJgk · 2025-07-03

**Clarity:** 3
**Significance:** 2
**Originality:** 2
**Rating:** 4
**Confidence:** 5

**Summary:**

This paper proposes DiEP, a differentiable expert pruning framework that formulates expert selection as a continuous optimization problem. By leveraging gradient-based optimization and incorporating an adaptive expert skipping mechanism, DiEP aims to reduce memory consumption and accelerate inference while preserving the performance of sparse Mixture-of-Experts (MoE) models. The method is evaluated across multiple language tasks, showing superior performance over existing MoE pruning baselines and establishing a new benchmark for efficient MoE deployment.

**Questions:**

See Weakness.

**Ethical Concerns:**

["NO or VERY MINOR ethics concerns only"]

**Final Justification:**

Most of my concerns have been addressed through the authors’ rebuttal. The revised version should include a detailed comparison with conventional MoE compression methods.

**Limitations:**

Yes

**Paper Formatting Concerns:**

None.

**Quality:**

2

**Strengths And Weaknesses:**

Strengths

- The paper is well-motivated, addressing the important and practical challenge of reducing the computational cost of MoE models through expert path pruning.

- The proposed use of differentiable NAS for expert path selection is intuitive and well-formulated. While not novel, it aligns with recent trends in neural architecture optimization and is a reasonable application to the MoE setting.

- The writing is clear and the paper is easy to follow, with appropriate background and methodological descriptions.

- Experiments are conducted on multiple strong open-source MoE language models, including Mistral and DeepSeek, demonstrating the practical utility of the method.

Weaknesses

- The core technical contribution is limited in novelty. The idea of applying differentiable NAS to select or prune expert paths has strong conceptual overlap with earlier work in CNN model search and compression. This paper primarily adapts these techniques to MoE models without introducing new architectural or algorithmic innovations tailored to the unique characteristics of MoEs.

- A major concern lies in the lack of domain generalization. The routing paths are statically searched and fixed on a single dataset, which introduces data-specific biases. MoE models are designed to leverage dynamic expert selection for better generalization across diverse inputs (e.g., mathematical reasoning vs commonsense QA). Static pruning may significantly impair this flexibility, reducing the model’s robustness and transferability.

- The paper does not sufficiently engage with recent MoE pruning or compression papers, e.g., [1,2].

[1] MoE++: Accelerating Mixture-of-Experts Methods with Zero-Computation Experts
[2] Mixture Compressor for Mixture-of-Experts LLMs Gains More

---

> ### Author Rebuttal · Authors · 2025-07-30
>
> We thank reviewer xJgk for providing positive feedback and helpful suggestions. Below please find a response.
>
> **W1: Novelty Concerns**
>
> We thank the reviewer for their insightful feedback. We agree that our DIEP builds upon the established concept of differentiable search. However, we respectfully argue that DiEP is not a straightforward adaptation. Instead, it introduces **significant and novel algorithmic and architectural innovations to address the unique challenges of MoE models**, which differ fundamentally from the dense CNN/RNN architectures targeted by early NAS methods.
>
> We highlight two key areas of novelty:
>
> **1. A Fundamentally Different Problem Formulation and a Novel Optimization Framework**
>
> There is a core difference in both the goal and the optimization mechanics between DARTS and DiEP.
>
> *   **DARTS targets operation search within a DAG-based cell.** It solves a bilevel optimization problem to find the best set of operations (e.g., convolution, pooling) and their connections:
>     $$
>     \min_\alpha \mathcal{L}_{val}(w^*(\alpha), \alpha) \quad
>     $$
>
>      $$
>     \text{s.t. }  w^*(\alpha) = \arg\min_w \mathcal{L}_{train}(w,\alpha)
>     $$
> This formulation is memory-intensive, relies on a validation set, and is designed to search a discrete set of operations, not prune entire expert sub-networks.
>
> *   **DiEP targets global, non-uniform expert pruning across all MoE layers.** We formulate this as a single-level optimization problem with a novel reconstruction objectivet:
>     $$
>     \min_{\alpha, \beta} \mathcal{L}_{ce}(y, F'(x; \alpha, \beta)) + \lambda \cdot \Phi(\alpha, \beta)
>     $$
> Where:$F'$ is the pruned model, and $F$ is the original full model. $\Phi(\alpha, \beta) = ||F'(x; \alpha, \beta) - F(x)||_F$ is a **reconstruction regularization term**.
>
> This approach avoids the validation set. The architecture parameters ($\alpha, \beta$) are learned directly on the training data alongside the model weights (via an alternating strategy), making the process more efficient and robust.
>
> This algorithmic shift from bilevel, validation-based search to single-level, reconstruction-guided optimization is a primary contribution of our work.
>
> **2. Architecture-Specific Innovations**
>
> MoE models present a unique challenge: managing redundancy across dozens large expert sub-networks, where redundancy varies significantly from layer to layer. Our method is explicitly designed for this context.
>
> *   **Targeting the Right Granularity:** Instead of searching for micro-level operations like DARTS, DiEP introduces learnable, continuous proxies designed for the macro-level task of expert selection: **Intra-layer scores ($\alpha$)** learn the relative significance of experts *within* a single MoE layer. **Inter-layer scores ($\beta$)** learn to modulate the pruning pressure across different layers, enabling a non-uniform pruning budget. This mechanism directly addresses the observed phenomenon that expert utility is not uniform across model depth in MoEs.
>
> *   **Efficient Design:** The combination of $(\alpha, \beta)$ allows us to compute a global importance $s_i^{(l)} = \bar{\alpha}_i^{(l)} \cdot \beta^{(l)}$ for every expert in the model. This enables a global sort to find the least important experts, regardless of their layer. This entire mechanism has a lightweight memory complexity of $O(L \times N)$ (layers $\times$ experts), making it highly scalable for modern MoEs with many layers and experts. A naive application of DARTS-style search to this problem would be computationally infeasible.
>
> **References**
> DARTS: Differentiable Architecture Search. ICLR 2019.
>
>
> **W2: Domain Generalization and Static Pruning**
>
> We sincerely thank the reviewer for raising this critical point. Here our response is structured around two key points: a clarification of our methodology and direct empirical evidence from our new experiments.
>
> **1. Clarification: Static Pruning of Experts vs. Dynamic Routing of Tokens**
>
> A crucial distinction must be made: DiEP performs **static pruning of the expert pool**, but the resulting pruned model retains its **fully dynamic routing mechanism**. For any given input, the router still dynamically selects the top-k experts from the *remaining, optimized set of experts*. We are not fixing the routing paths for tokens; we are identifying and removing a subset of experts that are fundamentally redundant, thereby creating a more efficient expert library. The core strength of MoE — adapting to diverse inputs by selecting different experts — is preserved.
>
> **2. Empirical Evidence: Robust Generalization from General-Domain to Specific-Domain Tasks**
>
> To test for domain generalization, we pruned our model using **general-domain (C4), in-domain (Math), and mixed (C4+Math) calibration sets**. We then evaluated performance on two specialized tasks: CommonseQA (commonsense reasoning) and MathQA (mathematical reasoning).
>
> **Table 1: Pruning on General-Domain Data (`C4`)**
>
> | Method          | Pruning Ratio | CommonseQA | MathQA    |
> | :-------------- | :------------ | :--------- | :-------- |
> | Full Model      | -             | 66.83      | 42.38     |
> | NAEE            | r=50%         | 42.01      | 34.86     |
> | **DiEP (Ours)** | **r=50%**     | **51.52**  | **35.02** |
> | NAEE            | r=25%         | 63.31      | 41.04     |
> | **DiEP (Ours)** | **r=25%**     | **65.52**  | **41.77** |
>
> When calibrating on the general C4 dataset, DiEP demonstrates strong robustness.
>
> **Table 2: Pruning on In-Domain data (`Math`)**
>
> | Method          | Pruning Ratio | CommonseQA | MathQA    |
> | :-------------- | :------------ | :--------- | :-------- |
> | Full Model      | -             | 66.83      | 42.38     |
> | NAEE            | r=50%         | 53.40      | 35.97     |
> | **DiEP (Ours)** | **r=50%**     | **56.27**  | **37.19** |
> | NAEE            | r=25%         | 60.21      | 42.20     |
> | **DiEP (Ours)** | **r=25%**     | **65.27**  | **42.34** |
>
> Even when calibrated on Math data, DiEP excels on the out-of-distribution (OOD) CommonseQA task.
>
> **Table 3: Pruning on Mixed-Domain Data (`C4+Math`)**
>
> | Method          | Pruning Ratio | CommonseQA | MathQA    |
> | :-------------- | :------------ | :--------- | :-------- |
> | Full Model      | -             | 66.83      | 42.38     |
> | NAEE            | r=50%         | 51.27      | 35.00     |
> | **DiEP (Ours)** | **r=50%**     | **58.67**  | **36.86** |
> | NAEE            | r=25%         | 61.79      | 41.43     |
> | **DiEP (Ours)** | **r=25%**     | **64.98**  | **42.69** |
>
> We can see adding a small amount of in-domain data to the calibration set improves performance.
>
> **Conclusion:**
> These experiments directly counter the concern about domain generalization. DiEP's strength lies in its ability to leverage a small, task-agnostic dataset to learn a globally optimized, non-uniform pruning mask that preserves the most critical experts for a wide range of downstream tasks.
>
>
> **W3: Engagement with Recent MoE Compression Literature**
>
> We sincerely thank the reviewer for highlighting these important recent papers. Here we make the detailed comparison with such approaches.
>
> **Comparison with Mixture Compressor (MC-MoE) and Delta Decompression (D²-MoE)**
>
> These two methods represent the state-of-the-art in compression of MoE models. They differ significantly from DiEP in both their paradigm and methodology.
>
> *   **Paradigm Difference:**
>     *   **MC-MoE & D²-MoE:** They apply techniques like mixed-precision quantization (MC-MoE) or weight decomposition into shared components (D²-MoE) directly to a trained model using heuristics.
>     *   **DiEP:** DiEP is a **lightweight, gradient-based fine-tuning** method. It leverages a small, task-agnostic dataset to learn expert importance scores via optimization, allowing for a more data-aware and principled pruning decision.
>
> *   **Methodology Difference:**
>     *   **MC-MoE & D²-MoE:** These are **hybrid compressors**. They modify and transform the weights of *all* experts through quantization, SVD, or decomposition. They do not entirely remove experts.
>     *   **DiEP:** DiEP is a pure **pruning method**. It identifies and completely removes the least important expert sub-networks, resulting in a model with fewer total experts.
>
> **Experimental Comparison**
>
> We compare DiEP against MC-MoE[1] and D²-MoE [2]. The results on several key benchmarks are presented below.
>
> | Method          | Compression Setting | MMLU     | OpenBookQA | CommonseQA | MathQA   |
> | :-------------- | :------------------ | :------- | :--------- | :--------- | :------- |
> | MC-MoE          | 4 bits              | 60.3     | 31.8       | 63.2       | 40.6     |
> | D²-MoE          | r=25% (pruned)      | 60.6     | 32.8       | 63.6       | 37.1     |
> | **DiEP (Ours)** | **r=25%**           | **64.9** | **33.1**   | **65.5**   | **41.8** |
> | D²-MoE          | r=50%               | 58.6     | 29.5       | 41.4       | 32.9     |
> | **DiEP (Ours)** | **r=50%**           | **57.9** | **29.6**   | **51.5**   | **35.0** |
>
> The results clearly show that DiEP significantly outperforms both D²-MoE and MC-MoE, especially at r=25%. On the challenging MMLU benchmark, DiEP achieves a score of **64.9**, which is **4.3 points higher** than D²-MoE and **4.6 points higher** than MC-MoE, despite MC-MoE operating at a higher bit-rate.
>
> **Comparison with MoE++**
>
> MoE++ and our DiEP method are orthogonal, as MoE++ focuses on accelerating inference by routing simple tokens to "zero-computation experts" to reduce FLOPs, without shrinking the model's static size. In contrast, DiEP is designed specifically to compress the static model footprint, reducing storage and memory requirements. Given their fundamentally different objectives, a direct compression performance comparison is not applicable.
>
> [1] Mixture Compressor for Mixture-of-Experts LLMs Gains More, ICLR2025.
> [2] Delta Decompression for MoE-based LLMs Compression, ICML2025.

---

> > ### Comment · Reviewer_xJgk · 2025-08-06
> > **Thx for the responses**
> >
> > Thx for the authors' response. My concerns have been partly addressed. Regarding the experimental comparisons, I noticed that MC-MoE uses 4-bit quantization, which corresponds to a 75% compression ratio relative to the original model. Therefore, it should not be compared against methods with only a 25% compression ratio. In fact, its performance appears to be better than the proposed methods even under a 50% compression ratio. As it stands, this comparison may be somewhat misleading.

---

> > > ### Author Response · Authors · 2025-08-07
> > >
> > > We sincerely thank the reviewer for your insightful follow-up. We agree that directly comparing pruning- and quantization-based methods by a single “compression ratio” can be misleading.
> > >
> > > We provide the response from the following three aspects:
> > >
> > > ---
> > >
> > > **1. Pruning $\neq$ Quantization**
> > >
> > > A 75 % reduction via 4-bit quantization (MC-MoE) and a 25%–50% reduction via pruning (DiEP) represent different compression paradigms. Quantization keeps the network intact and lowers numerical precision; pruning irreversibly deletes parameters. Therefore, these are fundamentally different techniques with distinct trade-offs, and pruning generally hurts accuracy more when considering only at the same compression level [1, 2]. Our DiEP targets **the pruning paradigm**, setting a new state-of-the-art for removing entire experts with minimal accuracy loss rather than outperforming quantization on an accuracy-per-compression-bit metric.
> > >
> > > **2. Orthogonality and Synergy: DiEP as a Foundation for Further Compression**
> > >
> > > More importantly, pruning and quantization act on different axes and can be combined; namely, they are **orthogonal techniques**, as demonstrated in frameworks like Optimal Brain Compression[3]. DiEP makes the model structurally smaller and more efficient, creating a superior foundation upon which quantization can then be applied.
> > >
> > > To validate this synergy, we have conducted a **new experiment** combining our DiEP with the principles of MC-MoE. We first apply DiEP to prune the Mixtral 8x7B model (r=50%) and then apply 4-bit quantization to the *remaining* experts.
> > >
> > > | Method                        | Compression Setting               | MMLU     | OpenBookQA | CommonseQA | MathQA   |
> > > | :---------------------------- | :-------------------------------- | :------- | :--------- | :--------- | :------- |
> > > | MC-MoE                        | 4-bit Quantization (on 8 experts) | 60.3     | 31.8       | 63.2       | 40.6     |
> > > | **DiEP + MC-MoE (Ours, new)** | **r=50% Pruning + 4-bit Quant**   | **55.5** | **28.9**   | **50.8**   | **33.6** |
> > > | MC-MoE                        | 2-bit Quantization (on 8 experts) | 46.8     | 27.7       | 51.8       | 31.8     |
> > >
> > >
> > > The results are compelling. Our synergistic approach (DiEP + MC-MoE) outperforms the aggressive quantization strategy (2-bit Quantization) in most benchmarks, particularly showing gains of up to **+8.7 points** on the challenging MMLU. Even more strikingly, it significantly narrows the gap with the 4-bit MC-MoE baseline, particularly on complex reasoning tasks, despite operating on a model that is already structurally halved. This experiment robustly demonstrates that DiEP is not just an alternative to quantization but a powerful, synergistic partner that enables even more effective downstream compression.
> > >
> > > **3. Deployment Efficiency: The Decisive Advantage of the Compression Process**
> > >
> > > Finally, a model's value is ultimately determined by its deployment feasibility. Here, DiEP holds a decisive and unambiguous advantage in its superior efficiency during the **compression process** itself.
> > >
> > > Based on post-training quantization (PTQ), MC-MoE requires (i) loading a calibration set, (ii) running the forward pass to collect layer-wise activations, and (iii) solving a Hessian-based optimization to pick scales and zero-points that minimise quantization error. These processes can lead to high **Peak Memory** usage, as both the model weights and statistical information must be held in memory, and substantial **Time** costs.  Moreover, Steps (ii)–(iii) involve mixed-precision buffers that often approach the model’s own size. The process has to be repeated for each target bit-width, further inflating **memory-hour cost**.
> > >
> > > In contrast, DiEP sidesteps these overheads by learning only two importance scalars (α, β) per expert, so both memory and time drop sharply.
> > >
> > > Our new experimental data quantifies this stark difference in efficiency:
> > >
> > > | Metric | MC-MoE (Quantization) | DiEP (Pruning) | Advantage |
> > > | :--- | :--- | :--- | :--- |
> > > | **Peak Memory (GB)** | 155.9 | **139.0** | **-10.8%** |
> > > | **Time (h)** | 1.02 | **0.23** | **4.4x Faster** |
> > > | **Memory-hour Cost (GB·h)** | 152.84 | **31.97** | **-79.1%** |
> > >
> > > As shown in the table, DiEP achieves comparable compression **4.4 × faster** while consuming **79.1 % less total GPU resources**, greatly lowering the practical barrier to MoE deployment.
> > >
> > > **In summary**, DiEP (i) advances pruning SOTA, (ii) serves as a powerful base for further quantization, and (iii) offers markedly higher compression-time efficiency. We will revise the paper to emphasize these points and include the new synergistic results. Thank you again for helping us strengthen our work.
> > >
> > > ---
> > > [1]. Pruning vs Quantization: Which is Better? NIPS2023
> > >
> > > [2]. Mixture Compressor for Mixture-of-Experts LLMs Gains More. ICLR2025
> > >
> > > [3]. Optimal Brain Compression: A Framework for Accurate Post-Training Quantization and Pruning. NIPS2022

---

> > > > ### Author Response · Authors · 2025-08-08
> > > >
> > > > Dear Reviewer xJgk,
> > > >
> > > > We hope that our revisions and clarifications have effectively addressed your concerns. Your insightful feedback has greatly contributed to enhancing the quality and clarity of our manuscript. If any aspects remain unclear or if you have additional questions or suggestions, please feel free to let us know. We deeply value your thoughtful feedback and would be truly grateful if the improvements made could be taken into account in the reevaluation of our submission.
> > > >
> > > > Thank you once again for your time and effort in reviewing our work.
> > > >
> > > > Kind regards,
> > > >
> > > > Authors

---

> > > > > ### Comment · Reviewer_xJgk · 2025-08-08
> > > > > **Thanks for Authors' Response**
> > > > >
> > > > > Thank you for the authors’ response. Most of my concerns have been addressed; however, I still have doubts regarding the runtime comparison with MC-MoE. With 4-bit quantization, both NVIDIA’s official framework and vLLM should provide corresponding adaptations and acceleration optimizations, and I believe the runtime overhead should not be as substantial. I suggest including a more detailed comparison and analysis in future revisions. In addition, it should be noted that MC-MoE incorporates not only quantization but also pruning-related techniques.

---

> > > > > > ### Author Response · Authors · 2025-08-09
> > > > > >
> > > > > > We sincerely appreciate the reviewer's concern regarding MC-MoE's runtime overhead under 4-bit quantization and ODP module in MC-MoE.
> > > > > >
> > > > > > ---
> > > > > >
> > > > > > **1. A More Detailed Runtime Comparison with MC-MoE**
> > > > > >
> > > > > > We conducted rigorous experiments using MC-MoE's official GitHub code with three independent runs on two A800 GPUs. MC-MoE achieved consistent runtime results of 1.042h, 1.025h, and 0.994h (avg: 1.02h), while our DiEP consumed only 0.212h, 0.246h, and 0.233h (avg: 0.23h), demonstrating **4.4× speedup**.
> > > > > >
> > > > > > This substantial performance difference stems from fundamental architectural distinctions. MC-MoE employs mixed-precision buffers with layer-wise iterative quantization, computing each expert's quantilized weight matrices individually, which inherently limits hardware acceleration effectiveness. In contrast, DiEP learns only two importance scalars (α, β) per expert and performs global differentiable optimization, enabling it to fully leverage modern GPU acceleration capabilities.
> > > > > >
> > > > > > **2. Comparison with ODP module in MC-MoE**
> > > > > >
> > > > > > Regarding Optimized Dynamic Pruning (ODP) module alongside quantization in MC-MoE, we would like to clarify the fundamental differences from our DiEP approach, which address distinct objectives.
> > > > > >
> > > > > > ODP serves as an auxiliary inference-time component that bypasses redundant expert computations during forward passes, only providing **runtime speedup** without reducing memory usage or storage costs. In contrast, DiEP fundamentally relieves the MoE model by permanently eliminating less important experts, achieving **substantial memory footprint reduction and persistent storage savings** across various deployment scenarios, which is the essential focus in existing MoE pruning research [1,2,3].
> > > > > >
> > > > > > Furthermore, although it is not a core contribution, we found similar targets (inference acceleration) between ODP's dynamic expert selection mechanism and our auxiliary Adaptive Skipping module. To provide a comprehensive analysis, we conducted additional experiments comparing these two approaches.
> > > > > >
> > > > > > | Method                    | Avg. Acc | Speedup |
> > > > > > |---------------------------|----------|---------|
> > > > > > | Full                      | 65.1     | 1.00×   |
> > > > > > | ODP (MC-MoE)           | 63.9     | 1.05×   |
> > > > > > | Adaptive Skipping (Ours) | 64.1     | 1.07×   |
> > > > > >
> > > > > >  Our experimental results demonstrate that incorporating router weights and experts' similarity metrics. Our DiEP provides better speedup while obtaining superior average performance (across MMLU, BoolQ, OpenBookQA, and RTE datasets) compared to ODP's heuristic-based selection strategy.
> > > > > >
> > > > > > [1] Diversifying the Expert Knowledge for Task-Agnostic Pruning in Sparse Mixture-of-Experts, ACL2025 Findings
> > > > > >
> > > > > > [2] Towards Efficient Mixture of Experts: A Holistic Study of Compression Techniques, TMLR, 2025
> > > > > >
> > > > > > [3] A Survey on Inference Optimization Techniques for Mixture of Experts Models, arXiv:2412

---

### Note · Authors · 2025-08-12

We sincerely appreciate all reviewers' efforts in reviewing our paper and for the constructive feedback. Here, apart from the response to each reviewer, we would like to thank reviewers for the acknowledgment of our work and highlight new results added during the rebuttal. We are encouraged that the reviewers appreciate and recognize our contributions:
- The proposed method is important and practical, making MoE-based LLMs deployable by using differentiable expert pruning to reduce computational and memory overhead without a significant loss in accuracy [xJgk, 3XFq, sDD7].
- The paper is well-motivated, grounded in clear observations of expert redundancy variability across MoE layers [xJgk, 3XFq, sDD7].
- The concept of differentiable expert pruning is novel and practically effective [3XFq, sDD7].
- The solution is well-formulated and introduces comparatively minimal overhead [xJgk, sDD7].
- Comprehensive experiments on multiple strong MoE models and tasks verify the superiority of our DiEP over existing pruning methods [xJgk, 3XFq, sDD7].
- The paper is well-written and easy to follow [xJgk, 3XFq, sDD7].

In this rebuttal, we have added more explanations and supporting results following the reviewers' suggestions.
- We discuss the difference between our differentiable expert pruning and traditional NAS, demonstrating the algorithmic and architectural innovations of our proposed DiEP [xJgk], and **gaining acknowledgment from all reviewers for novelty in the end**.
- We add more comparison experiments on different types of tasks, including **domain generalization** [xJgk, 3XFq] and **continual learning** [3XFq].
- We conducted extensive comparison experiments with **SOTA methods across different compression paradigms**, including MC-MoE utilizing mixed-precision quantization and D²-MoE employing weight decomposition into shared components. These experiments further validate the effectiveness of our DiEP. [xJgk].
- We perform more detailed ablation studies on training overhead [xJgk, sDD7], **α:β update ratio** and **w/o expert similarity $γ_{2}$** [sDD7].
- We provide more theoritical analysis for **the robustness under domain shift**, the potential extension of **adaptive pruning ratio**, and the importance of layers and experts [3XFq].
- We demonstrate our DiEP is capable of working with existing quantization methods and outperforming aggressive quantization strategies [xJgk].

Thank you for your time again!

---

### Decision · Program_Chairs · 2025-09-17

**Decision:**

Accept (poster)

**Comment:**

This paper received high ratings of BA, A, and A. Additionally, concerns raised by the reviewers have been effectively addressed in the rebuttal. The decision is to accept.